# Climatic control of the surface mass balance of the Patagonian Icefields

Tomás Carrasco-Escaff[1,2], Maisa Rojas[1,2], René Garreaud[1,2], Deniz Bozkurt[2,3], and Marius Schaefer[4]

[1]Department of Geophysics, University of Chile, Santiago, Chile
[2]Center for Climate and Resilience Research, University of Chile, Santiago, Chile
[3]Department of Meteorology, University of Valparaíso, Valparaíso, Chile
[4]Instituto de Ciencias Físicas y Matemáticas, Universidad Austral de Chile, Valdivia, Chile

**Correspondence:** Tomás Carrasco-Escaff (tcarrasco@dgf.uchile.cl)

**Abstract.** The Patagonian Icefields (Northern and Southern Patagonian Icefield) are the largest ice masses in the Andes Cordillera. Despite its importance, little is known about the main mechanisms that underpin the interaction between these ice masses and climate. Furthermore, the nature of large-scale climatic control over the surface mass variations of the Patagonian Icefields still remains unclear. The main aim of this study is to understand the present-day climatic control of the surface mass balance (SMB) of the Patagonian Icefields at interannual timescales, especially considering large-scale processes.

We modeled the present-day (1980-2015) glacioclimatic surface conditions for the southern Andes Cordillera by statistically downscaling the output from a regional climate model (RegCMv4) from a 10 km spatial resolution to a 450 m resolution grid, and then using the downscaled fields as input for a simplified SMB model. Series of spatially averaged modeled fields over the Patagonian Icefields were used to derive regression and correlation maps against fields of climate variables from the ERA-Interim reanalysis.

Years of relatively high SMB are associated with the establishment of an anomalous low-pressure center near the Drake Passage, the Drake low, that induces an anomalous cyclonic circulation accompanied with enhanced westerlies impinging the Patagonian Icefields, which in turn leads to increases in the precipitation and the accumulation over the icefields. Also, the Drake low is thermodynamically maintained by a core of cold air that tends to reduce the ablation. Years of relatively low SMB are associated with the opposite conditions.

We found low dependence of the SMB on main atmospheric modes of variability (El Niño-Southern Oscillation, Southern Annular Mode), revealing a poor ability of the associated indices to reproduce the interannual variability of the SMB. Instead, this study highlights the Drake Passage as a key region that has the potential to influence the SMB variability of the Patagonian Icefields.

## 1 Introduction

The Patagonian Icefields (Northern Patagonian Icefield (NPI) and Southern Patagonian Icefield (SPI)) are the most extensive ice bodies in the Andes Cordillera. Given their size, they play a significant role in modulating the local and regional environment, providing ecosystem processes such as climate regulation, gas regulation, and hydrologic cycles regulation, among others

(Martínez-Harms and Gajardo, 2008; Dussaillant et al., 2012). Both icefields have been losing mass over the last decades (Rignot et al., 2003; Malz et al., 2018; Minowa et al., 2021), and recent evidence shows that they are the primary contributors to sea-level rise among all South American ice masses (Braun et al., 2019; Dussaillant et al., 2019). Overall, glaciers of the Southern Andes have contributed approximately 3.3 mm of sea-level rise between 1961 and 2016 (Zemp et al., 2019). Despite the importance of the Patagonian Icefields, little is known about the main mechanisms underpinning the interaction between these ice bodies and climate, especially the large-scale climate processes that determine their surface mass balance (SMB) at interannual timescales. This topic represents a significant issue for understanding the Patagonian Icefields' past, present, and future evolution and, more generally, the southern Andean cryosphere.

The Patagonian Icefields spread over the 46-52ºS (Fig. 1), a latitudinal band influenced by the continuous passage of mid-latitude systems embedded in an intense westerly flow (Trenberth, 1991; Berbery and Vera, 1996; Hoskins and Hodges, 2005). The steep north-south oriented topography induces a substantial orographic enhancement of precipitation on the windward side and a rain-shadow effect on the leeward side (Roe, 2005; Jobbágy et al., 1995; Garreaud et al., 2009). This generates a temperate and hyper humid climate to the west of the Andean ridge and an arid and continental climate eastward (Paruelo et al., 1998; Carrasco et al., 2002; Aravena and Luckman, 2009; Garreaud et al., 2013). Several authors reported a regional warming trend in Patagonia (Rosenblüth et al., 1997; Rasmussen et al., 2007; Olivares-Contreras et al., 2019), whereas regional precipitation trends are spatially inhomogeneous (Quintana and Aceituno, 2012; Aravena and Luckman, 2009; Garreaud et al., 2013). For example, Boisier et al. (2018) report negative trends to the north of the NPI, while González-Reyes et al. (2017) find positive trends to the south of the SPI.

The Patagonian climate is controlled primarily by the strength of the westerly winds (Garreaud et al., 2013). Garreaud et al. (2013) find a high correlation between zonal wind and precipitation in western Patagonia at daily, monthly, and interannual timescales. They also find a seasonal correlation between zonal wind and temperature, indicating that windy summers tend to be colder than average and windy winters tend to be warmer than average. Consequently, modes of variability affecting the westerly flow impact the Patagonian climate profoundly, such as the Southern Annular Mode (SAM), the leading mode of extratropical Southern Hemisphere variability (Fogt and Marshall, 2020, for a review). This mode is characterized by an equivalent barotropic, zonally symmetric structure involving exchanges of mass between the mid and high latitudes with positive polarity associated with a strengthening and poleward shifting of the polar jet and negative polarity associated with a weakening and equatorward shift of the polar jet (Rogers and Van Loon, 1982; Thompson and Wallace, 2000). Additionally, subsidence and adiabatic warming occur in the troposphere on the equatorward side of the polar jet during the positive phase of SAM, while opposite temperature anomalies maintain during the negative phase (Fogt and Marshall, 2020).

The circumpolar anomalies in westerly flow and tropospheric temperature exhibited during each phase of SAM lead to corresponding anomalies in precipitation and surface temperature in Patagonia. In particular, southern South America (south of 40ºS) exhibits warmer than average conditions during the positive phase of SAM, while opposite anomalies maintain during the negative phase (Garreaud et al., 2009). In terms of precipitation, during the positive phase of SAM, northern Patagonia exhibits drier than average conditions, and southern Patagonia exhibits moister than average conditions, while the opposite occurs during the negative phase (Garreaud et al., 2009). During the last decades, a southward shift and strengthening of the

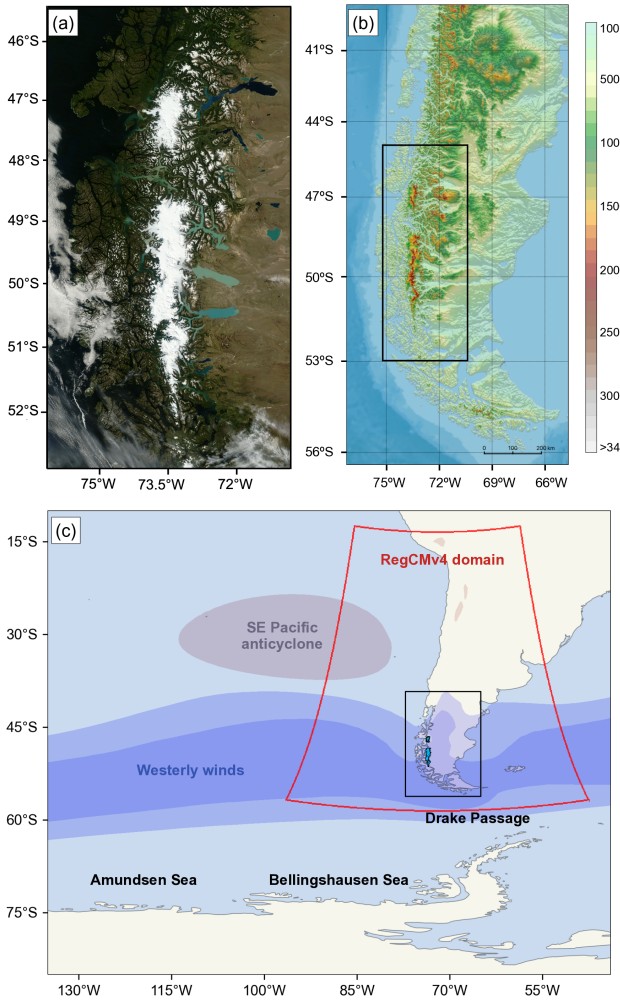

**Figure 1.** (a) Satellite image of Northern Patagonian Icefield and Southern Patagonian Icefield taken by the MODIS sensor on board the NASA's TERRA satellite on February 19, 2011. (b) Terrain elevation (m a.s.l.) of southern South America obtained from the digital elevation model ETOPO1 with 1 minute of arc resolution. The black box spans the area of panel (a). (c) Schematic of the main features of large-scale circulation near the Patagonian Icefields. The red polygon indicates the spatial domain used for running the RegCMv4 present-climate simulations. The black box spans the area of panel (b).

Southern Hemisphere westerly wind belt has been observed (e.g., Goyal et al., 2021), and consistently SAM has shown a significant positive trend associated primarily with ozone depletion and increase of greenhouse gases (Gillett and Thompson, 2003; Arblaster and Meehl, 2006). This trend has favored dry conditions in northern Patagonia, mainly during austral summer (Boisier et al., 2018), and moist conditions in southern Patagonia (González-Reyes et al., 2017). In turn, these moister than average conditions in southern Patagonia have been suggested to significantly influence the SMB of ice bodies to the south of the SPI (Möller et al., 2007).

Further modulation of the Patagonian climate is due to the El Niño-Southern Oscillation (ENSO), the Earth's largest source of year-to-year climate variability (Wang et al., 2017, for a review). During ENSO events, stationary Rossby wave trains are generated in response to deep convection generated by tropical sea surface temperature (SST) anomalies (Hoskins and Karoly, 1981; Karoly, 1989). These wave trains, identified in the Southern Hemisphere with the Pacific-South American pattern (Mo and Higgins, 1998; Mo and Paegle, 2001), include anomalous anticyclonic circulation over the Amundsen–Bellingshausen Seas in the southeastern Pacific (Karoly, 1989) and are associated with enhanced blocking episodes in this region (Rutllant and Fuenzalida, 1991; Jacques-Coper et al., 2016; Demortier et al., 2021). These circulation anomalies can substantially influence the precipitation regime of southern South America. For instance, they have been related to a decrease in precipitation in western Patagonia during ENSO warm (El Niño) events, especially during summer, with the opposite conditions during cold (La Niña) events (Montecinos and Aceituno, 2003; Schneider and Gies, 2004; Weidemann et al., 2018a; Garreaud, 2018; Agosta et al., 2020).

Furthermore, Cai et al. (2020) found slightly different spatial patterns of precipitation anomalies for Central Pacific ENSO events and Eastern Pacific ENSO events (Capotondi et al., 2015; Timmermann et al., 2018). The diversity of spatial patterns and intensities of SST anomalies in the tropical Pacific Ocean among ENSO events results in different atmospheric circulation responses (Taschetto et al., 2020), which in turn would affect the linkage between ENSO and Patagonian climate. Thus, even though drier and warmer than normal conditions are expected during El Niño events, especially during summer, the net effect of ENSO on the Patagonian climate seems to depend on the specifics of each ENSO event.

The meteorological conditions over the Patagonian Icefields have a direct impact on the glaciological surface processes (e.g., snowfall, surface melting) that determine the gain (accumulation) and loss (ablation) of mass experienced by the glaciers. The SMB corresponds to the overall sum of the surface accumulation and surface ablation, i.e., the net change of mass at the surface, over a certain period of time. Unlike the total mass balance or the glacier geometry, the SMB integrates the direct interplay between glaciers and climate and thus represents a suitable study variable for assessing climate-cryosphere interaction.

Similar to many mountain regions in the world, the Patagonian Icefields show a lack of in situ climatic and glaciologic measurements due to difficult access and harsh environmental conditions. There are few measurement stations and short records available (Fig. S1), which hinder a robust assessment of glacier response to current climate conditions. Due to the inadequate observational network, various studies have tried to quantify the SMB of the Patagonian Icefields using different global gridded climate datasets (i.e., reanalysis), downscaling techniques (dynamical and statistical downscaling procedures), and SMB models of different complexity (Schaefer et al., 2013, 2015; Lenaerts et al., 2014; Mernild et al., 2017). Interestingly, all studies found positive trends in the SMB of the Patagonian Icefields and a positive SMB for the SPI. Nonetheless, none of them assesses the interannual variability of the SMB nor its relationship with local and large-scale atmospheric processes.

On the other hand, several studies were published in the last two decades in which the mass loss of the Patagonian Icefield was quantified by remote sensing methods (Rignot et al., 2003; Willis et al., 2012a, b; Jaber et al., 2016; Malz et al., 2018; Foresta et al., 2018; Braun et al., 2019; Dussaillant et al., 2019; Minowa et al., 2021). The mass loss shown by these works is especially strong over the SPI (e.g., Braun et al., 2019; Dussaillant et al., 2019), which contrasts with the aforementioned positive SMB and highlights the importance of frontal ablation in total mass balance calculations. The methodological approach

followed by these studies does not allow us to obtain a quantitative assessment of the SMB variability and its relationship with atmospheric processes. Thus, there is a substantial gap in understanding how atmospheric processes affect the SMB of the Patagonian Icefields, especially at interannual timescales.

Motivated by improving our knowledge about the climate-cryosphere interplay, this paper links the annual anomalies in the SMB of the Patagonian Icefields with local, regional, and large-scale climate anomalies. The main goal of this work is to understand the present-day climatic control of the SMB of the Patagonian Icefields at interannual timescales, especially considering large-scale processes. Understanding the mechanisms behind year-to-year changes in the SMB is an essential requirement for deepening the comprehension of the climate processes responsible for past, present, and future trends of the SMB of the Patagonian Icefields and an important opportunity for future development of diagnostic and prognostic tools.

To achieve our goal, we first simulate present-day glacioclimatic surface conditions for the southern Andes Cordillera using a simplified SMB model forced with a high-resolution regional climate model simulation. Then, we average the modeled fields over the Patagonian Icefields and compute the time series of anomalies of the SMB in order to derive regression and correlation maps against fields of climate variables. In this work, we seek to obtain a robust estimation of the interannual variability of the SMB of the Patagonian Icefields rather than getting exact estimates for the mean values of the modeled variables. Thus, we devote effort to analyzing the sensitivity of the modeled SMB interannual variability to several modeling aspects (e.g., main model parameters and the mean value of the input fields). The paper is structured as follows: in Sect. 2, we describe the study area, data, and methods used in the study. In Sect. 3 and Sect. 4, we present the results and discussion, respectively. Finally, in Sect. 5, we present the conclusions of our work.

## 2 Study Area, Data, and Methodology

### 2.1 Study Area

The study area comprises the Patagonian Icefields. They spread over a latitudinal band of 46-52º S and include the NPI and the SPI (Fig. 1). The NPI locates between 46º30' S and 47º30' S and covers a total ice area of 3953 $km^2$ (Rivera et al., 2007). It elongates in the north-south direction with an axis near the 73º30' W, extending ∼100 km in length and 40-45 km in width (Aniya, 1988). It shows a steep topography with terrain elevation values increasing eastward in most parts of the icefield area, reaching the sea level at the west margin and a maximum of 3970 m a.s.l. (above sea level) at the summit of Mount San Valentín. Characteristic terrain elevation values are 1000 m a.s.l. for the west side and 1500 m a.s.l. for the east side (Warren and Sugden, 1993). The NPI is composed of 38 glaciers larger than 0.5 $km^2$ (Dussaillant et al., 2018).

The SPI locates between 48º20' S and 51º30' S and covers a total ice area of 12514 $km^2$ (Casassa et al., 2014). It extends ∼350 km in length and generally 30-40 km in width, with the narrowest part only 8 km wide (Aniya et al., 1998). This icefield contains a central plateau lying between the 1400-2000 m a.s.l. with terrain elevation values decreasing southward. The SPI reaches its topographic maximum at Volcán Lautaro with a peak of 3607 m a.s.l. It is composed of 48 main outlet glaciers (Aniya et al., 1998).

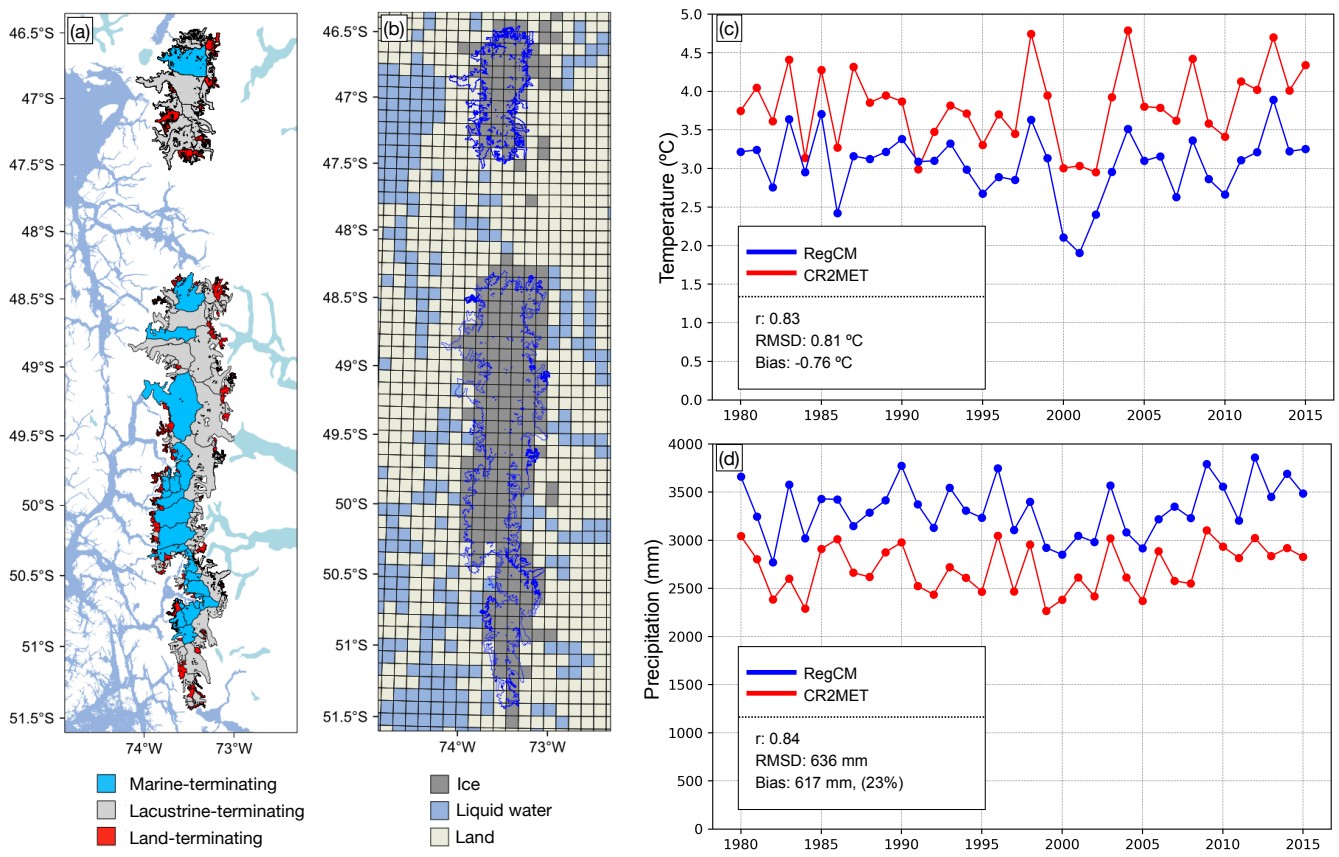

**Figure 2.** (a) Patagonian Icefields together with their glacier divides and type of terminus. (b) The RegCMv4 grid (∼10 km spatial resolution), the model land use (grid box colors) and the NPI and SPI outlines (blue contours). (c) Comparison of annual mean temperature time series (box at 46-52º S and 72.5-74.5º W) using RegCMv4 data and CR2MET data. (d) Same as (c) but for accumulated annual precipitation.

## 2.2 Data

Simulated meteorological fields of near-surface air temperature, precipitation, and surface downward solar radiation were obtained from the regional climate model RegCM version 4.6 (hereafter RegCMv4) at 10 km spatial resolution and 3 h

temporal resolution for the period 1980-2015 (Bozkurt et al., 2019). The regional climate simulation setting consisted of two nested domains at 0.44º (∼50 km) and 0.09º (∼10 km) spatial resolutions and 23 sigma levels. Initial and boundary conditions for the mother domain were provided by the European Centre for Medium-Range Weather Forecasts (ECMWF) Reanalysis (ERA-Interim) at 6 h temporal resolution and 0.75º spatial resolution, including SST fields. More information about the RegCMv4 and simulations can be found in Bozkurt et al. (2019).

RegCMv4 forced by ERA-Interim simulations were evaluated in previous studies (Bravo et al., 2019, 2021). Using observed accumulated snowfall observation from the ultrasonic-depth gauges located in northern SPI, Bravo et al. (2019) demonstrated

that the estimated accumulated snowfall values using different phase partitioning methods were in the range of the observed values (see their Fig. 7). Nonetheless, it is also important to mention that some inherent uncertainties and errors may exist in the simulated fields of SMB components such as precipitation, which are mainly associated with the boundary conditions and physical configuration used in the model (e.g., radiation and cumulus schemes).

Due to the scarcity of in-situ measurements, we were unable to directly validate the RegCMv4 fields against observations for the spatial and temporal scales involved in this study. Instead, we verified the RegCMv4 near-surface temperature and precipitation fields using the high-resolution gridded meteorological dataset CR2MET (Alvarez-Garreton et al., 2018). CR2MET has a spatial resolution of 0.05 degree and we used v1.4.2 for precipitation and v1.3 for temperature, depending on the availability of data. The CR2MET dataset is based on in situ observations of precipitation and temperature for the territory of continental Chile, covering the period 1979–present. This dataset is partly based on a statistical downscaling of ERA-Interim reanalysis. In terms of precipitation, the dataset considers the local topography, which is defined by a set of calibrated parameters with local rainfall observations. Similarly, land surface temperature estimates from MODIS satellite retrievals are considered in the statistical downscaling approach and near-surface temperature provided by ERA-Interim. More detailed information about the CR2MET can be found in Alvarez-Garreton et al. (2018).

A brief evaluation of the CR2MET dataset is given in Figs. S2-S4. Using the available surface meteorological stations in the region of interest for the period 1980-2015, CR2MET tends to have a wet bias in the annual precipitation (4.9%) and a cold bias in the annual mean daily minimum and maximum temperatures (-0.4C and -0.6C, respectively). Nonetheless, CR2MET overall reproduces reasonably well the interannual variability of the aforementioned variables, which is consistent with the main purpose of this study. Because of the absence of solar radiation fields in the CR2MET dataset, and in order to maintain physical coherence in the fields employed, we preferred to use the RegCMv4 fields as the meteorological forcing of the SMB model, and use the CR2MET for verification purposes.

Elevation terrain data were obtained from the NASA Shuttle Radar Topography Mission Global 3 arc second sub-sampled (STRMGL3S V003; hereafter STRMv3) distributed by NASA Making Earth System Data Records for Use in Research Environment (MEaSUREs) SRTM (NASA JPL, 2013), which has a 3" horizontal resolution ($\sim$90 m). Additionally, glacier extent data were obtained from the Randolph Glacier Inventory Version 6 (RGIv6; RGI Consortium, 2017). The RGIv6 is a globally complete inventory of glacial outlines. It has a collection of vector data that describes the geometry associated with various glaciers and other types of information such as area, mean elevation, and type of term. It is a supplement to the GLIMS initiative (Global Land Ice Measurements from Space) that aims to be a photograph of world-wide glacier extent at the beginning of the 21st century.

To assess large scale patterns associated with SMB anomalies, data from several climatic variables for the period between 1980-2015 were taken from the ERA-Interim dataset with a grid spacing of 0.75º × 0.75º (Dee et al., 2011) including surface air temperature (SAT), zonal and meridional wind (u and v, respectively), mean sea level pressure (MSLP), geopotential height (Z), and SST. Also, outgoing longwave radiation (OLR) flux at the top of the atmosphere data was taken from the NOAA Climate Data Record (CDR) Program, with a grid spacing of 2.5º x 2.5º (Lee and NOAA CDR Program, 2011).

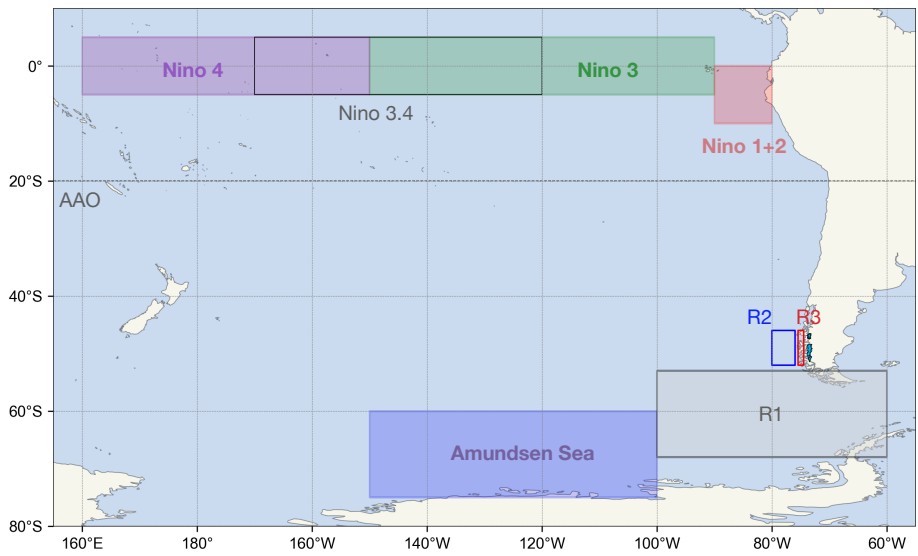

**Figure 3.** Regions used for the construction of climate indices.

To characterize ENSO we used the monthly SST averaged over the Nino1+2 and the Nino3.4 regions (see Fig. 3) obtained from the NOAA Climate Prediction Center (CPC, https://psl.noaa.gov/data/climateindices/list/). Additionally, we used the Central Pacific (CP) and Eastern Pacific (EP) ENSO indices to account for ENSO diversity (Kao and Yu, 2009; Yu et al., 2012). To obtain the spatiotemporal pattern of the EP ENSO, SST anomalies regressed with the Nino4 index were removed from total SST anomalies before performing Empirical Orthogonal Function (EOF) analysis. The same approach is used for computing the CP index but using the Nino1+2 index instead. To characterize SAM activity we used the AAO index obtained from the NOAA CPC. We used monthly means of daily values from 1980 to 2015. The daily AAO index was constructed by projecting the daily 700 hPa height anomalies poleward of 20°S onto the leading mode of the EOF analysis performed on the monthly mean 700 hPa height. This leading mode was obtained using the period 1979-2000.

We constructed custom indices of climatic variables averaged over specific regions of interest (Fig. 3). We spatially averaged the monthly values of the ERA-Interim geopotential height at 300 hPa and air temperature at 850 hPa in a box near the Drake Passage, spanning the 68-53º S in latitude and 100-60º W in longitude (box R1 in Fig. 3). We did the same with the southeast Pacific SST next to central Patagonia (box R2 in Fig. 3, at 52-46º S and 80-76º W) and the zonal wind at 850 hPa impinging central Patagonia (box R3 in Fig. 3, at 52-46º S and 75.5-74.5º W). We named these time series Z300 Drake, T850 Drake, SST-R2, and U850-R3, respectively.

### 2.3 Methodology

To overcome the lack of observational surface data in the area, we modeled present-day glacioclimatic surface conditions for the southern Andes cordillera. First, we statistically downscaled near-surface air temperature, precipitation, and surface downward solar radiation fields obtained from RegCMv4 to a 450 m resolution grid. Then, we used these meteorological fields

as input for a simplified SMB model. Later, we derived the time series of the modeled fields by computing the spatially averaged meteorological forcing and the glaciological output fields over the Patagonian Icefields area (assigning the same weight to each grid point). Only grid points within a mask file of the Patagonian Icefields were used for spatial-average comparisons. After that, we computed the annual, winter and summer time series of the modeled fields using hydrological years from April to March, winters from April to September, and summers from October to March. Finally, we used the time series of annual SMB anomalies to perform correlation and linear regression analysis with several climatic variables.

It is also important to mention that since we are interested in obtaining a robust estimation of the interannual variability of the SMB of the Patagonian Icefields, rather than getting exact estimates for the mean values, we performed several sensitivity experiments to test the dependence of the modeled SMB interannual variability on the main modeling aspects.

### 2.3.1 Statistical downscaling of the RegCMv4 output

The RegCMv4 Digital Elevation Model (DEM) tends to underestimate the terrain elevation when compared with the SRTMv3 DEM, especially at higher elevations, both in the NPI and the SPI (Fig. S5). In order to avoid biases in near-surface temperature and precipitation due to elevation biases, we corrected the near-surface temperature and precipitation RegCMv4 model output accounting for the biases in the RegCMv4 DEM.

To do so, we first constructed a DEM resulting from the averaged SRTMv3 DEM at every five grid points ($\sim$450 m spatial resolution) and used this as the default model DEM. This spatial resolution was selected after performing a sensitivity analysis (Carrasco-Escaff, 2021). We then statistically downscaled the RegCMv4 main surface atmospheric output (near-surface air temperature, precipitation, and surface downward solar radiation) from 10 km spatial resolution to the 450 m resolution grid. Regarding the temporal resolution, each field remained at a 3 h resolution.

The statistically downscaling process started with the bilinear interpolation of the fields onto the RegCMv4 DEM to remap data from a 10 km resolution to a 450 m grid resolution of the default model DEM. Then, we performed altitudinal corrections for temperature and precipitation. In the case of temperature, we applied a constant lapse rate equal to the environmental lapse rate (6.5 ℃ km$^{-1}$). In this way, we computed the near-surface temperature every 3 h according to:

$$T = T_{\text{bil}} - \text{LR} \cdot (z - z_{\text{bil}}) \tag{1}$$

where $T$ is the downscaled near-surface temperature, $T_{\text{bil}}$ is the 450 m bilinearly interpolated RegCMv4 near-surface temperature, LR = 6.5 ℃ km$^{-1}$ is the lapse rate, $z$ is the model reference DEM, and $z_{\text{bil}}$ is the 450 m bilinearly interpolated RegCMv4 DEM. In the case of precipitation, we used the following equation at every 3 h:

$$P = P_{\text{bil}} \cdot (1 + \text{PG} \cdot (z - z_{\text{bil}})) \tag{2}$$

where $P$ is the statistically downscaled precipitation, $P_{\text{bil}}$ is the 450 m bilinearly interpolated RegCMv4 precipitation, and PG = 0.05% m$^{-1}$ is the precipitation gradient (as in Schaefer et al., 2013, 2015). Finally, we remapped the original RegCMv4 surface downward solar radiation to the 450 m grid by performing bilinear interpolation.

### 2.3.2 SMB model

We used the statistically downscaled fields as input for a simplified SMB model. The SMB model output consists of accumulation, ablation, and SMB fields with 3 h temporal resolution and 450 m spatial resolution. The accumulation ($c$) is modeled using two temperature thresholds to determine the proportion of precipitation that falls as snow (e.g. Schaefer et al., 2013, 2015; Bravo et al., 2019). At every grid cell, the fraction $q$ of precipitation that falls as snow is determined by the near-surface temperature ($T$) of the grid cell and calculated according to:

$$q = \begin{cases} 0, & \text{if } T_{\text{th}} \leq T \\ \frac{T_{\text{th}} - T}{T_{\text{th}}}, & \text{if } 0\ ^{\circ}\text{C} \leq T < T_{\text{th}} \\ 1, & \text{if } T < 0\ ^{\circ}\text{C} \end{cases} \tag{3}$$

where $T_{\text{th}}$ is the temperature threshold at which precipitation falls as rain. This parameter is set to 2 °C and the sensitivity of model results to this choice is tested below. Accumulation is defined as the solid part of precipitation ($P$) and computed as:

$$c = q \cdot P \tag{4}$$

The ablation ($a$) at every grid cell is represented by melting, and it is computed using a simplified energy balance model in which the sum of longwave radiation and turbulent fluxes is approximated by a linear function in temperature (Oerlemans, 2001). First, the surface energy flux ($\psi$) is calculated according to:

$$\psi = (1 - \alpha) \cdot R + c_0 + c_1 \cdot T \tag{5}$$

where $\alpha$ is the surface albedo, $R$ is the surface downward solar radiation, $T$ is the near-surface temperature in °C, and $c_0$ and $c_1$ are the calibration parameters. Then, the ablation is computed as:

$$a = \begin{cases} \frac{\psi}{L_m \rho_w} \cdot \Delta t, & \text{if } \psi > 0 \\ 0, & \text{if } \psi \leq 0 \end{cases} \tag{6}$$

where $L_m = 333.55 \times 10^{-3}$ J kg$^{-1}$ is the latent heat of fusion, $\rho_w = 1000$ kg m$^{-3}$ is the liquid water density and $\Delta t = 10800$ s. The SMB model assigns one of three types of surface to every grid cell: snow, firn, and ice. Each type of surface has a specific albedo (snow albedo is 0.85, firn albedo is 0.55, and ice albedo is 0.35). The albedo values were taken from Cuffey and Paterson (2010) and correspond to the recommended values for fresh dry snow, clean firn and clean ice. In the SMB model, every grid cell consists of a column of ice at the bottom, possibly followed by a column of firn and possibly by a column of snow. At every 3 h and for each grid cell, the SMB model calculates the accumulation added to the column of snow. Then, the SMB model computes the ablation, and the model simulates the melting of the (possible) snow, followed by the (possible) firn and the ice. Finally, the SMB is computed at every 3 h and for each grid cell according to:

$$b = c - a \tag{7}$$

At the start of each autumn season (1 April), the mass of firn in the firn column turns into ice, and the mass of snow in the snow column turns into firn. Initially, each grid cell consists of only a column of ice (infinitely deep), and the SMB model was forced with the downscaled climatological conditions obtained from the RegCMv4 for five years before feeding it with the actual RegCMv4 downscaled fields.

### 2.3.3 Calibration

The parameters $c_0$ and $c_1$ were calibrated using the SMB estimations for the NPI and the SPI from Minowa et al. (2021). These are the only known estimates based on an observational approach covering both the NPI and the SPI in an icefield-wide sense. Minowa et al. (2021) estimated a -1.5 Gt yr$^{-1}$ SMB annual rate for the NPI in the period 2000-2019 and an 11.5 Gt yr$^{-1}$ SMB annual rate for the SPI in the same period. Even though the temporal coverage of our SMB model ends in 2015, we calibrated the parameters $c_0$ and $c_1$ for the period 2000-2015 to compare with the values reported by Minowa et al. (2021).

To calibrate the model parameters, we ranged $c_0$ from -48.0 to 48.0 Wm$^{-2}$ every 1.0 Wm$^{-2}$ and $c_1$ from 9.5 to 11.5 Wm$^{-2}$°C$^{-1}$ every 0.5 Wm$^{-2}$°C$^{-1}$. For every pair of values $(c_0, c_1)$, we computed the annual rate of SMB for the NPI and the SPI between 1 April 2000 and 31 March 2015. Then, we compared it with the estimates from Minowa et al. (2021) for the period 2000-2019. The closest value for the NPI was found using the calibration parameters $c_0 = -6.0$ Wm$^{-2}$ and $c_1 = 9.5$ Wm$^{-2}$°C$^{-1}$, producing an SMB annual rate of -1.48 Gt yr$^{-1}$. Meanwhile, the closest value for the SPI was reached using the values $c_0 = 21.0$ Wm$^{-2}$ and $c_1 = 9.5$ Wm$^{-2}$°C$^{-1}$, giving an SMB annual rate of 11.41 Gt yr$^{-1}$.

### 2.3.4 Sensitivity analysis

The absence of observational data in the study area in an icefield-wide sense imposes a limitation on validating the SMB model output. To overcome this limitation, we conducted several sensitivity experiments to determine the dependence of the modeled SMB (especifically its interannual variability) on the main model parameters ($c_0$, $c_1$, the surface albedos and the threshold at which precipitation falls as rain), the complexity of the solar radiation remapping, and the mean value of the meteorological input. Additionally, we tested the degree of dependence of the modeled SMB on the interannual variability of each meteorological input. A suite of sensitivity simulations with the aforementioned aspects was used to obtain annual estimates of SMB, which were then compared with the control time series.

*Sensitivity to main model parameters.* To assess the sensitivity of the modeled SMB to $c_0$, we added an offset ($\Delta c_0$) to the parameter value ranging from -4.0 to 4.0 Wm$^{-2}$ every 1.0 Wm$^{-2}$. This offset was added to the $c_0$ calibrated value of each icefield simultaneously, and then the model was rerun and the output was compared against the original SMB. The sensitivity of the modeled SMB to the $c_1$ parameter was explored in a similar manner, varying an offset ($\Delta c_1$) from -2.0 to 2.0 Wm$^{-2}$°C$^{-1}$ every 0.5 Wm$^{-2}$°C$^{-1}$. To assess the sensitivity of the modeled SMB to the surface albedo, we replaced the original albedo parametrization (hereafter A0) with two additional parametrizations. The first parametrization (A1) uses albedo values of 0.75, 0.50, and 0.30 for snow, firn, and ice, respectively, and corresponds to minimum albedo values for fresh dry snow, clean firn and clean ice (Cuffey and Paterson, 2010). The second parametrization (A2) uses albedo values of 0.50, 0.30, and 0.20 for snow, firn and ice, respectively, and corresponds to recommended values for old debris-rich dry snow, debris-rich firn and debris-rich

ice (Cuffey and Paterson, 2010). Finally, the sensitivity of the modeled SMB to the threshold at which precipitation falls as rain was studied adding an offset ($\Delta T_{\text{th}}$) to this parameter ranging from -1.00 to 1.00 ºC every 0.25 ºC.

*Sensitivity to the complexity of the insolation remapping.* In the original time series of the modeled SMB, bilinear interpolation of the solar radiation does not account for terrain parameters such as slope and aspect, which can interfere with energy flux at the surface. Therefore, we also assessed our simple approach by comparing with an alternative downscaling technique

that considers terrain parameters. Briefly, we first calculated the solar radiation reaching the (sloped) surface under clear sky conditions ($R_{s,cs}$) for the SRTMv3 grid (only the NPI area) using the SRTMv3 DEM and a radiation code (Corripio, 2003). Then, we computed the solar radiation reaching the surface under clear sky conditions but assuming a horizontal surface instead ($R_{h,cs}$). Next, we used conservative remapping to upscale $R_{h,cs}$ to the RegCMv4 grid and calculated a cloud factor dividing the RegCMv4 surface downward solar radiation in the upscaled $R_{h,cs}$. After that, we performed bilinear interpolation to remap

the cloud factor field to the SRTMv3 grid and we computed the terrain-modified solar radiation as the product of the cloud factor and the $R_{s,cs}$ field. Finally, we used conservative remapping to upscale the terrain-modified solar radiation to the 450 m grid. With this solar radiation, we rerun the SMB model and compared the output with the original SMB.

*Sensitivity to the mean value of the meteorological input.* We examined whether possible biases in the mean state of the meteorological input fields could lead to potential differences in terms of the interannual variability of the SMB of the Patagonian

Icefields. A brief evaluation of the simulated mean annual climate by RegCMv4 with respect to CR2MET is given in Fig. 2c, d. Overall, it can be stated that the model well captures the interannual variability. Nonetheless, there exist systematic colder (bias of -0.76 ºC) and wetter estimates (bias of 23%) compared to CR2MET, which can be associated with biases of boundary conditions and the regional climate model itself (see also Bravo et al., 2019, 2021). Therefore, and taking into account the biases from the CR2MET dataset, we added an offset $\Delta T$ to the near-surface temperature ranging from -1.5 to 1.5 ºC every

0.5 ºC and weighted the precipitation with a factor $P_0$ ranging from 0.8 to 1.2 every 0.1. For each pair of values ($\Delta T, P_0$), we rerun the SMB model and compared with the original SMB.

Finally, we compared the RegCM4 surface downward solar radiation field with the upscaled field $R_{h,cs}$. For every grid cell in a region considering the NPI, if the daily RegCMv4 total cloud fraction was less than 1%, we calculated the ratio of daily energy flux at the surface between both fields. The resulting distribution indicated that the RegCMv4 systematically estimates

lower values of solar radiation reaching a flat surface under clear sky condition (mean error of -19% and lower quartile of -35%). Consequently, we weighted the solar radiation with a factor $R_0$ ranging from 1.1 to 1.5 every 0.1, rerun the model and compared the ouptut with the original SMB.

*Sensitivity to the interannual variability of the meterological input.* We studied the sensitivity of the modeled SMB to the interannual variability of the downscaled fields of temperature, precipitation, and insolation. To do this, we conducted experi-

ments in which we (i) removed the variability of a period $\tau \geq 1$ yr from a specific meteorological field (e.g., precipitation); (ii) remained the other meteorological fields unmodified (e.g., temperature and solar radiation); (iii) rerun the SMB model; and (iv) computed the time series of the spatially averaged field of SMB using the new output. In order to remove the variability of a period $\tau \geq 1$ yr from a specific meteorological field, we calculated, for each grid point, the annual cycle of that field (retaining the mean value) at a temporal resolution of 3 hr, and then we used that cycle repeatedly to feed the SMB model.

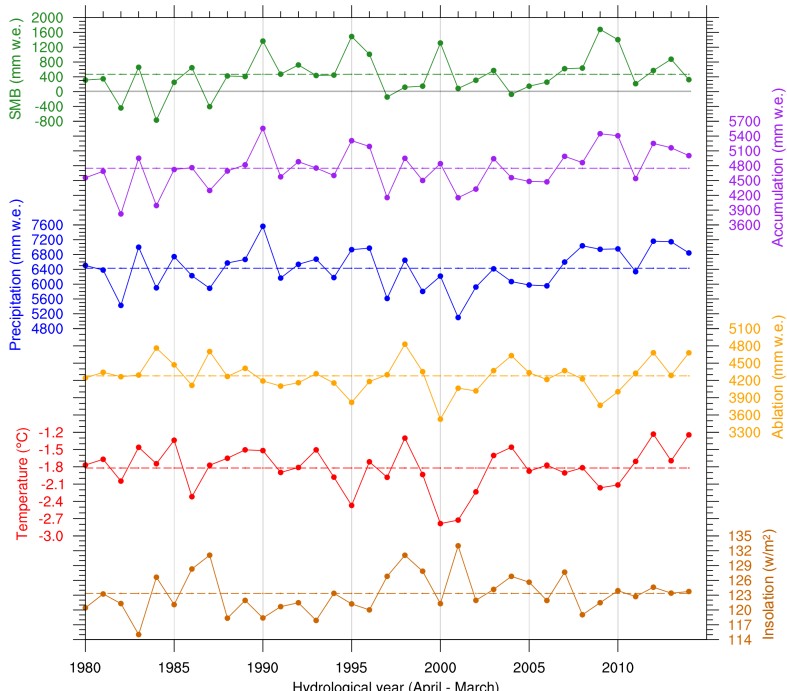

**Figure 4.** Annual (April to March) time series of spatially averaged (over the area of the Patagonian Icefields) fields of (from top to bottom) SMB, accumulation, accumulated precipitation, ablation, mean near-surface temperature and mean insolation. Each segmented line indicates the mean value of the series. The grey horizontal line corresponds to the SMB zero isoline.

We assessed the degree of dependence of the interannual variability of the SMB on the interannual variability of a specific meteorological variable $X$ by calculating the squared correlation ($R^2$) between the original SMB time series and the SMB time series computed from the corresponding experiment. A high value of $R^2$ indicates that a large part of the variance of the original series could be explained even with $X$ reduced to its annual cycle; thus, the interannual variability of the original SMB depends poorly on the year-to-year variations of $X$. In this way, we interpreted a high (low) value of $R^2$ as a low (high) degree of dependence in terms of interannual variability. Finally, we also calculated the squared correlation between the original SMB time series and the SMB time series computed from each experiment at winter and summer timescales.

## 3 Results

### 3.1 Mean values and covariability

Fig. 4 shows the annual time series of the modeled SMB, accumulation, ablation, precipitation, temperature, and insolation, calculated as the spatial average of each field over both icefields. The mean values of these time series are tabulated in Table S1 and their standard deviations in Table S2. The modeled annual SMB averages 469 $\pm$ 537 mm w.e. (mean $\pm$ std. dev.;

w.e. stands for water equivalent). During winter (April to September), the modeled SMB increases up to $1806 \pm 331$ mm w.e., while during summer (October to March) it decreases down to -1336 $\pm$ 428 mm w.e. The modeled annual precipitation averages $6430 \pm 536$ mm w.e., and about 74% of the precipitation falls as snow. On average, winter and summer precipitation values are very similar, yet winter accumulation is 1.24 times the summer accumulation. The modeled ablation shows a greater seasonal difference with a summer mean value of 4.17 times the winter mean value. The modeled mean annual temperature ($-1.82 \pm 0.37$ ºC) is below the freezing point, averaging -3.44 $\pm$ 0.57 ºC and -0.20 $\pm$ 0.42 ºC during winter and summer, respectively. Finally, the modeled annual insolation averages $123 \pm 4$ $\mathrm{Wm}^{-2}$ with a summer mean value of 3.43 times the winter mean value.

Correlation between pairs of time series was computed (Tables S3, S4, and S5), and hypothesis testing was performed at a significance level of 5% ("*" corresponds to statistically significant values). Results show that the annual SMB is highly and positively correlated with the annual accumulation ($r = 0.87^*$) and negatively correlated with the annual ablation ($r = -0.69^*$). During winter, the correlation between the SMB and the accumulation increases ($r = 0.94^*$), while the correlation between SMB and the ablation decreases and becomes statistically non-significant. During summer, the correlation between the SMB and the accumulation equals in magnitude the correlation between the SMB and the ablation ($r = -0.90^*$). Among the modeled meteorological variables, the annual SMB is found to have the largest correlation with the annual precipitation ($r = 0.69^*$), followed by annual insolation ($r = -0.44^*$) (see Table S3). The same order is also evident in winter. The correlation between the SMB and temperature is only significant in summer. Additionally, results show that the modeled meteorological variables are correlated with each other. For instance, the annual precipitation and insolation show a moderate but significant correlation ($r = -0.53^*$), while annual temperature and insolation show no significant correlation ($r = -0.19$) (Table S3). Annual temperature and precipitation are positively correlated, reaching a value of $r = 0.45^*$. During winter, this correlation increases ($r = 0.61^*$), while, during summer, it becomes almost null ($r = 0.05$).

In order to further examine the annual time series of the spatially averaged SMB field, we compared the time series with the leading mode of interannual variability of the SMB field. After conducting an EOF analysis, we retained the first three leading modes of variability following the North's rule of thumb (North et al., 1982). Results are shown in Figs. S6 and S7. The leading mode of variability explains 59% of the total variance and dominates most of the area of the Patagonian Icefields. This mode shows a high correlation (Pearson's r coefficient) with the annual SMB of the glaciers located at the western margin of the icefields, especially those situated on the SPI. On the other hand, it tends to become null at the ablation zone of glaciers located at the eastern margin of the icefields, suggesting that these zones behave independently from the rest of the icefields (Fig. S6d). The annual time series of the spatially averaged SMB field virtually coincides with the leading mode of interannual variability of the SMB field (Fig. S6a); thus, it well represents the interannual variability of the SMB for most of the grid points over the Patagonian Icefields.

## 3.2 Sensitivity analysis

The results of the sensitiviy experiments (see Sect. 2.3.4) reveal that the interannual variability of the SMB is insensitive to slight changes in the main model parameters, namely the calibration parameters of the ablation module, the albedo parametriza-

**Table 1.** Slope of the linear regression of the annual (April to March) time series of the spatially averaged SMB with the annual (April to March), winter (April to September) and summer (October to March) time series of the spatially averaged fields of SMB, accumulation, ablation, accumulated precipitation, mean temperature and mean insolation. (*) Statistically significant value at a significance level of 5%.

| Slope of linear regression | SMB | Accumulation | Ablation | Precipitation | Temperature | Insolation |
|---|---|---|---|---|---|---|
| Units | mm w.e./std. dev. | | | | °C/std. dev. | $Wm^{-2}$/std. dev. |
| Annual | 537* | 351* | −186* | 368* | −0.09 | −1.72* |
| Winter | 200* | 188* | −12 | 212* | 0.04 | −1.14* |
| Summer | 336* | 163* | −173* | 156* | −0.23* | −2.31* |

tion and the threshold at which precipitation falls as rain (Table S6). Analogously, the analysis shows that small biases in the mean state of the meteorological fields do not affect the overall interannual variability of the SMB (Tables S7 and S8). In each case, the annual SMB time series obtained after varying the model parameters or the meteorological input shows a high correlation with the original SMB, and both series feature similar standard deviations. Finally, replacing the bilinear interpola-

370 tion of the surface downward solar radiation by a downscaling technique considering incidence angles at each grid cell of the high-resolution topography (SRMTv3) lowers the mean insolation and increases the SMB but does not modify the interannual variability of the series (Table S6).

Additionally, we assessed the degree of dependence of the variability of the SMB on the variability of the meteorological variables (see Sect. 2.3.4, Table S9). The results show almost no dependence of the annual variations of the SMB on the annual

variations of insolation ($R^2 = 95\%$). Similar results are found when we analyze winter-to-winter variations ($R^2 = 99\%$) and summer-to-summer variations ($R^2 = 95\%$). The annual variations of SMB show a high dependence on the annual variation of temperature and precipitation, and this dependency is found to be larger on annual precipitation ($R^2 = 26\%$) compared to that on annual temperature ($R^2 = 45\%$). The same order of dependency is also evident in winter. Nonetheless, summer temperature variations appear to have larger influences on the variations of SMB than summer precipitation variations.

### 3.3 Local-scale control over the SMB

Previous results suggest that the local-scale control over the SMB is exerted primarily by the temperature and precipitation. Precipitation exerts the primary control at annual and winter timescales, and the temperature does it at the summer timescale. To perform further examination, we assessed the local-scale control over the SMB using regression analysis on the time series of spatially averaged fields over the Patagonian Icefields. These time series were regressed onto the annual SMB anomaly

time series. This allows us to estimate the characteristic variation of each modeled glaciological and meteorological variable associated with a positive or negative anomaly of annual SMB (measured in std. dev. units). The results of the regression analysis are shown in Table 1. We tabulated only the slope of each regression since the intercept corresponds to the mean value of each dependent variable (Table S1).

For simplicity, we analyze a year when the SMB is one standard deviation above the mean value, which corresponds to an annual SMB anomaly of 537 mm w.e. (the analysis extends linearly to other cases). Such a year is associated with an annual accumulation anomaly of 351 mm w.e. and an annual ablation anomaly of -186 mm w.e. Regarding seasonal differences, years with relatively large SMB are associated with higher than average winter and summer accumulation values and lower than average summer ablation values. The winter accumulation anomaly is 1.15 times the summer accumulation anomaly but the summer ablation anomaly is more than 14 times the winter ablation anomaly. As a result, when grouping contributions by process, the annual SMB anomalies are primarily explained by accumulation anomalies, while when grouping by season, the summer anomalies in the glaciological processes account for most of the annual SMB anomalies.

The same analysis (i.e., the SMB is one standard deviation above the mean value) yields an increase of winter precipitation of 212 mm w.e., whereas an almost null and not statistically significant variation in winter temperature. During summer, there is an increase of precipitation of 156 mm w.e. and a variation in temperature of -0.23 °C. Thus, our results suggest that years with higher than average SMB are related to wetter than normal annual conditions and colder than normal summer conditions, while years with lower than average SMB are associated with the opposite.

### 3.4 Regional-scale control over the SMB

To assess the regional scale control over the SMB, we first computed the regression of the annual SMB anomalies with the annual precipitation, near-surface temperature, and horizontal wind (at 10 m above ground level and 700 hPa). Results are shown in Fig. 5a, c. For simplicity, we analyze the years when the SMB is above the mean value (the analysis extends linearly to other cases). Positive anomalies of annual SMB are associated with an intensification of the westerly winds impinging the Austral Andes, a regional cooling in the south of South America and over the Pacific Ocean adjacent to Patagonia, and an increase (decrease) of the precipitation to the west (east) of the Andean ridge. The cooling is stronger over the Pacific Ocean adjacent to central and north Patagonia and northeast of the Patagonian Icefields. The increase in precipitation reaches the highest values in central-western Patagonia, with a maximum over the Patagonian Icefields. Over the Pacific Ocean adjacent to Patagonia, the circulation acquires anticyclonic vorticity to the north and cyclonic vorticity to the south, both at the near-surface level and at the 700 hPa pressure level. Some differences in horizontal wind anomalies are evident when comparing near-surface level and at 700 hPa pressure level due to the topographic blocking imposed by the Andes.

We also computed latitudinal profiles of regressions of the annual SMB with the mean annual fields of zonal wind, geopotential height, and air temperature at a longitude of 80 °W. Results are shown in Fig. 5b, d. The intensification of the westerly winds during years of relatively high SMB comprises the southern tip of the Andes, extending from near 38 °S to near 60 °S and maximizing near 50 °S. At altitude, the positive zonal wind anomaly extends throughout the entire troposphere and reaches its maximum around 300 hPa, between cores of high and low anomalous geopotential height, in a region where the pressure gradient is maximum. In turn, these cores of anomalous geopotential height are located in regions where the magnitude of the temperature gradient is maximum, resembling a thermal wind balance. Interestingly, the anomalous cold region below the core of low anomalous geopotential height extends to the lower troposphere and comprises the latitudinal band where the Patag-

**Annual SMB projected onto annual fields of selected variables**

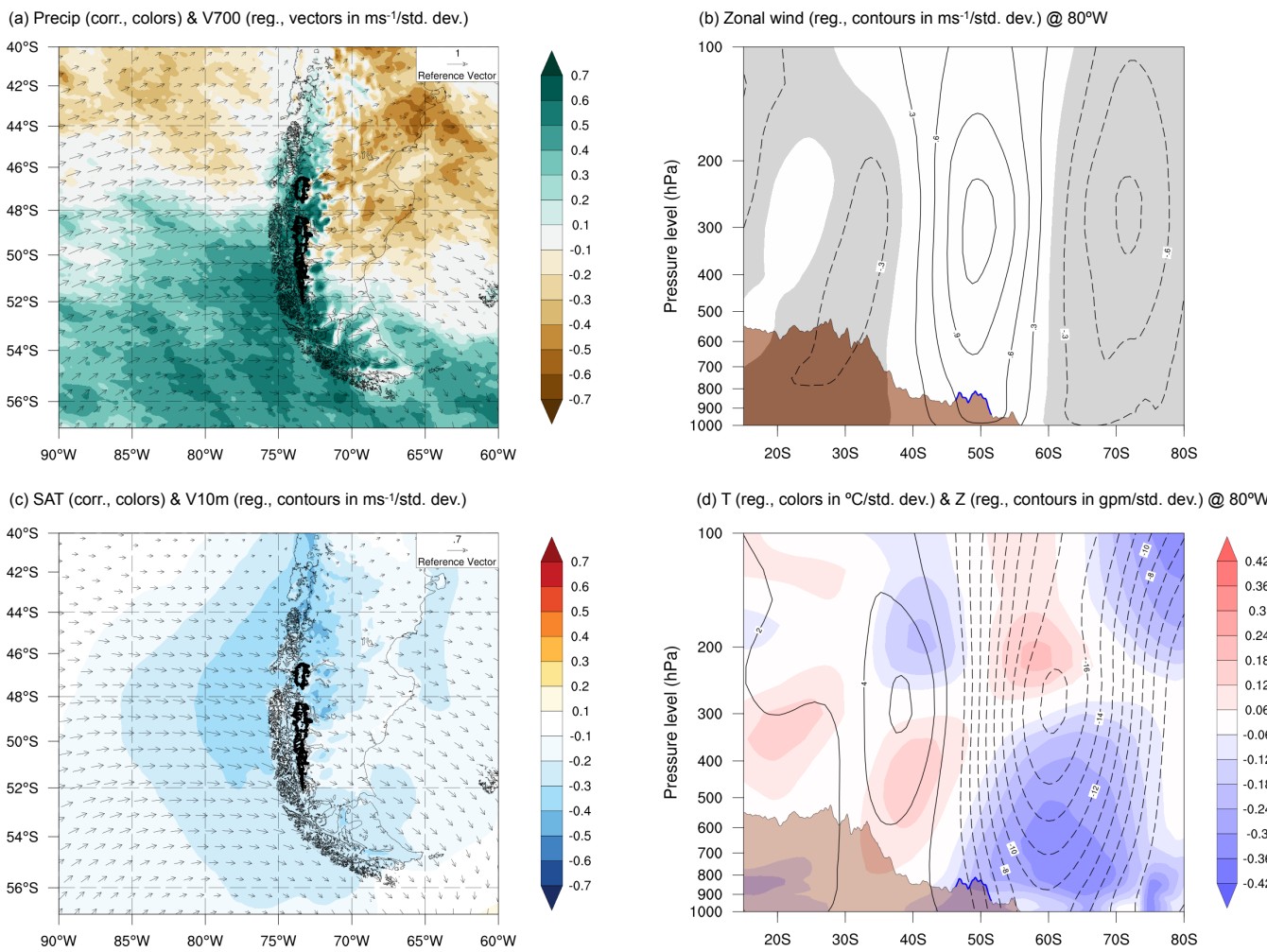

**Figure 5.** Regional correlation and linear regression maps of the annual (April to March) time series of the spatially averaged field of SMB with fields obtained from the RegCMv4 simulation data and ERA-Interim reanalysis. a) Regression with annual fields of horizontal wind at 700 hPa (vectors in $\text{ms}^{-1}$/std. dev.) and correlation with accumulated precipitation (colors). Fields were obtained from the RegCMv4 data. b) Latitudinal and atmospheric profile of the regression with annual field of zonal wind (contours in $\text{ms}^{-1}$/std. dev.) for a transect at 80 ºW. Negative regression values are shaded, and the Andes topography within the latitudinal band of the Patagonian Icefields is shown with blue lines. Fields were obtained from the ERA-Interim reanalysis. c) Regression with annual fields of horizontal wind at 10 m above ground level (vectors in $\text{ms}^{-1}$/std. dev.) and correlation with the mean near-surface air temperature (colors). Fields were obtained from the RegCMv4 data. d) Latitudinal and atmospheric profile of the regression with annual field of geopotential height (contours in gpm/std. dev.) and temperature (colors in ºC/std. dev.) for a transect at 80 ºW. The Andes topography within the latitudinal band of the Patagonian Icefields is shown with blue lines. Fields were obtained from the ERA-Interim reanalysis.

onian Icefields are located. This suggests that during years of relatively high SMB, the reinforcement of the westerlies wind impinging Patagonia and the temperature anomaly observed in the Patagonian Icefields could be linked to the same mechanism.

Regarding seasonal differences, years with SMB above the average show a stronger circulation and a more pronounced precipitation change during winter than summer (Figs. 6 and 7). Also, these years are associated with a pronounced summer cooling over the south of South America and the adjacent Pacific Ocean (Fig. 7c), while correlations with winter near-surface temperature are virtually null. The latitudinal profiles also show a stronger reinforcement of the westerly winds during winter than summer (Fig. 6b), associated with more pronounced cores of anomalous geopotential height (Fig. 6d). Nonetheless, during summer, the low-pressure structure appears displaced northward, especially in the lower troposphere (Fig. 7d). A more intense cooling of the anomalous cold region tends to concentrate in the lower troposphere (1000 to 700 hPa), which could explain the summer cooling observed along the Patagonian Icefields.

## 3.5   Large-scale control over the SMB

To assess the large-scale control over the SMB, we computed the regression fields of several climatic variables (at annual, winter, and summer timescales) onto the annual SMB anomaly time series. The results are shown in Figs. 8-10.

Years with SMB above the average are characterized by the presence of an anomalous low-pressure center located around the Drake Passage (hereafter Drake low) with a longitudinal extension from the northeastern Amundsen Sea and northeastern Antarctic Peninsula (∼120ºW to 50ºW), and a latitudinal extension from the west Antarctic coast to the southern tip of South America (Fig. 8a). Around the Drake low, anomalous high-pressure centers are established over the subtropical South Pacific, extending towards the Amundsen Sea and the South Atlantic.

The Drake low is associated with an anomalous cyclonic circulation established around the Drake Passage (Fig. 8b). A strengthening of the annual zonal winds in the latitudinal band comprising the Patagonian Icefields and the longitudinal band comprising the 60-120 ºW is observed, while a weakening of the zonal wind is exhibited southward. Furthermore, an intensification of the trade winds is also observed over the central equatorial Pacific, with magnitudes comparable to the ones exhibited by the westerly winds impinging the Patagonian Icefields.

Regarding SST anomalies (Fig. 8a), positive anomalies of annual SMB are associated with a surface cooling off the coast of Patagonia, in accordance with the regional-scale analysis (see Fig. 5). A large-scale cooling is observed around the central-eastern equatorial Pacific and the west coast and southern tip of South America, resembling an Eastern Pacific La Niña-like pattern (Fig. S10). Nonetheless, this pattern is latitudinally asymmetric with respect to the equator, and the strongest SST correlations are associated with off-equatorial tongues off the coast of South America. In addition, beneath the anomalous cold tongue around the equatorial Pacific, an anomalous warm tongue emerges from the western equatorial Pacific towards the subtropical South Pacific.

Years with SMB above the average are associated with positive OLR anomalies over the central equatorial Pacific (i.e., decreased convective activity) and negative OLR anomalies over the western equatorial Pacific (Fig. 8c), consistent with the SST patterns. These OLR anomalies are accompanied by anomalies of geopotential height at 300 hPa that account for both (i) an equivalent barotropic arrangement of the Drake low, which extends throughout the troposphere, and (ii) a series of low, high

**Annual SMB projected onto winter fields of selected variables**

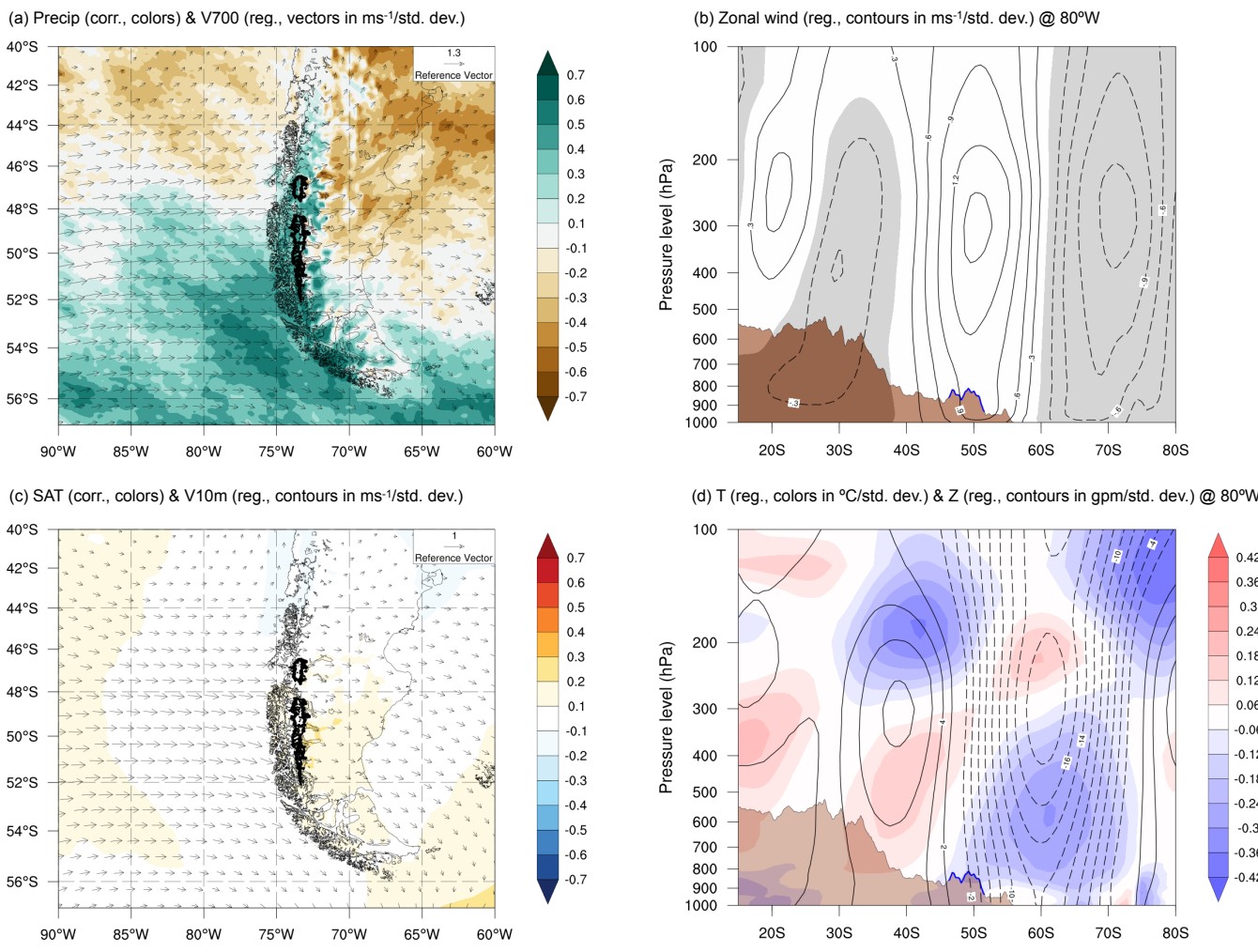

**Figure 6.** The same as in Fig. 5 but for the winter (April to September).

and low anomalies pressures that appear to spread from the tropics to the extratropics as a result of inhibition of convective activity over the central equatorial Pacific.

Concerning seasonal differences, years with SMB above the average are characterized by a deeper and more extensive Drake low during winter than summer and a more marked anomalous high-pressure signal to the north of the Patagonian Icefields as well (Figs. 9, 10). Moreover, a more pronounced enhancement of the westerly winds impinging Patagonia is evident during winter than summer, and the same occurs with the intensification of the trade winds in the central equatorial Pacific. Conversely, the cooling signal over the western Patagonian coasts and the equatorial Pacific Ocean is much more intense during summer

**Annual SMB projected onto summer fields of selected variables**

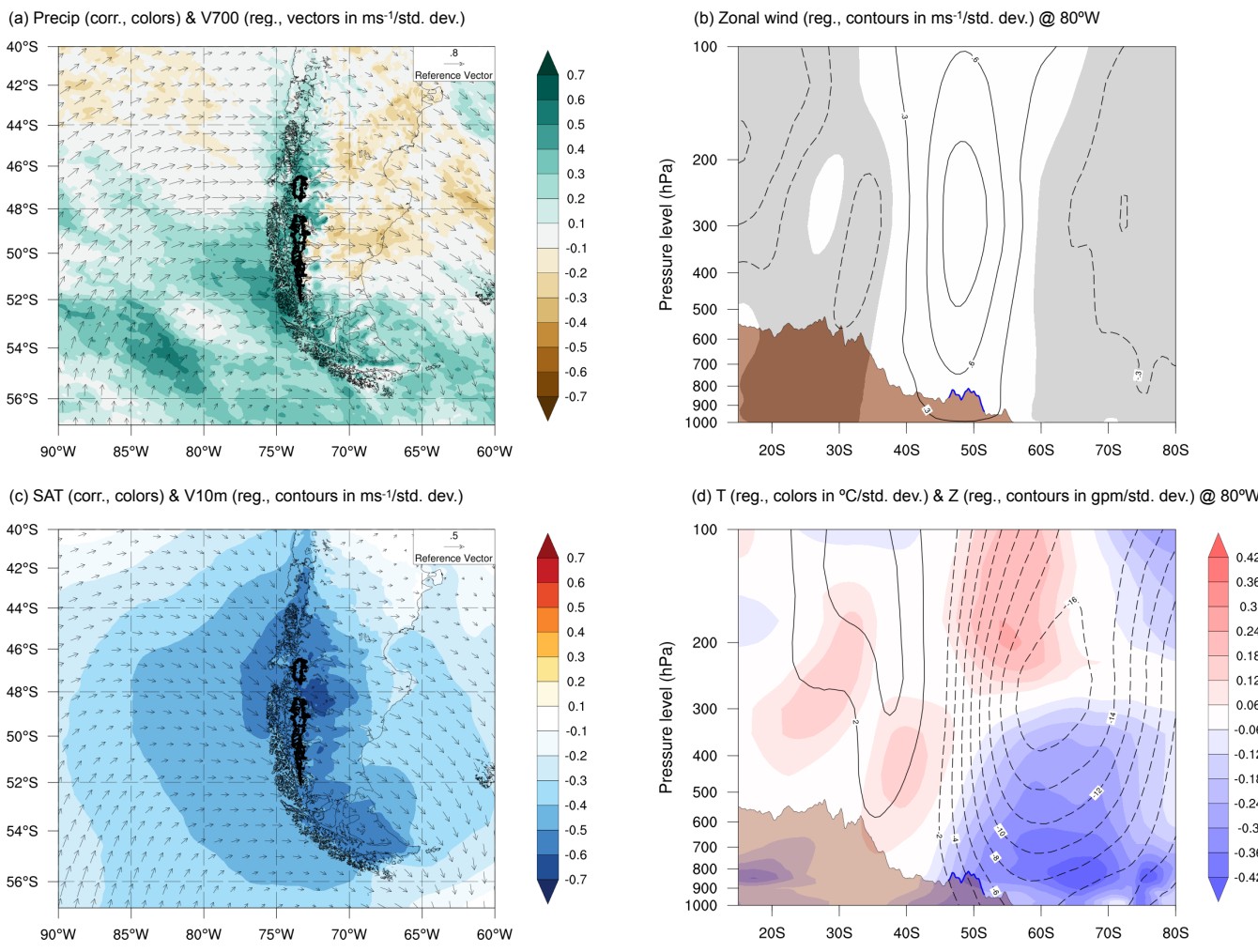

**Figure 7.** The same as in Fig. 5 but for the summer (October to March).

than the winter (Figs. 9a, 10a), and the OLR and geopotential height at 300 hPa anomalies are also much more evident during summer than winter (Figs. 9c, 10c).

**3.6 Correlation with large-scale indices**

Table 2 shows the correlation between indices of main modes of interannual variability affecting the Patagonian climate and the time series of spatially averaged fields of SMB, accumulated precipitation, and near-surface temperature at annual, winter, and summer timescales. Additionally, we show the correlation with the custom indices described in Sect. 2.2.

**Annual SMB projected onto annual fields of selected variables**

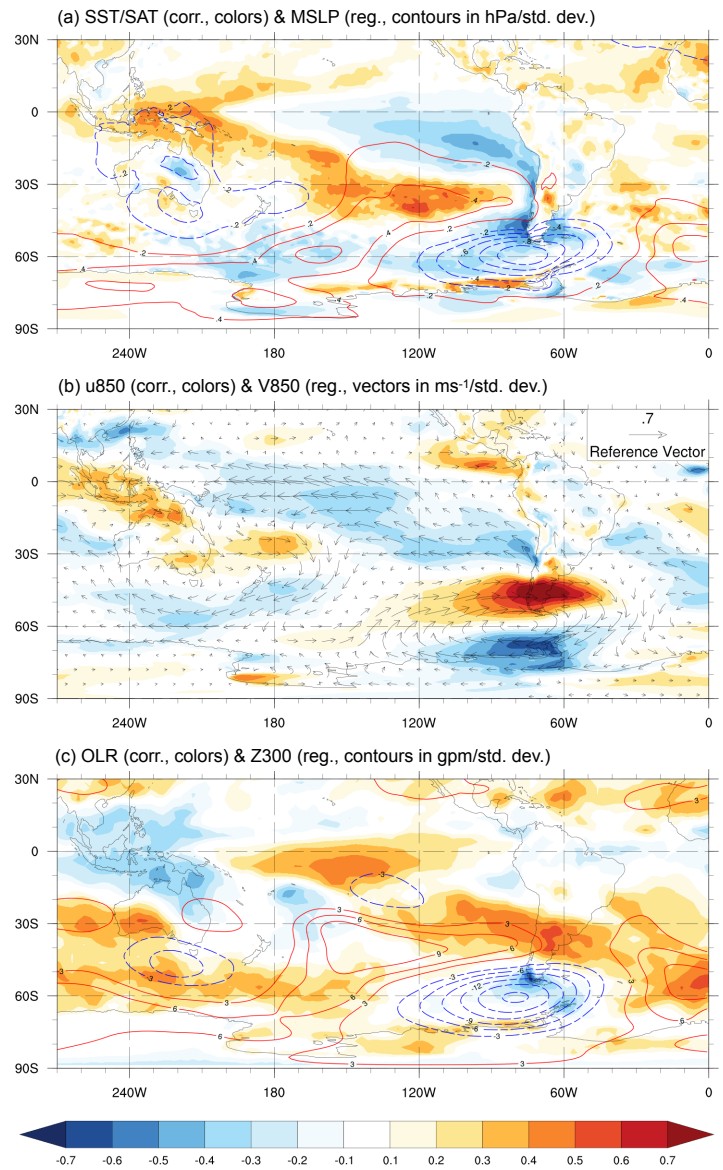

**Figure 8.** Large-scale correlation and linear regression maps of the annual (April to March) time series of the spatially averaged field of SMB with fields obtained from the Era-Interim reanalysis. a) Regression with the annual field of mean sea level pressure (contours in hPa/std. dev.) and correlation with the annual field of sea surface temperature (colors over ocean) and near-surface air temperature (colors over land). b) Regression with the annual field of horizontal wind at 850 hPa (vectors in $\mathrm{ms}^{-1}$/std. dev.) and correlation with the annual field of zonal wind at 850 hPa (colors). c) Regression with the annual field of geopotential height at 300 hPa (contours in gpm/std. dev.) and correlation with the annual field of outgoing longwave radiation (colors).

**Annual SMB projected onto winter fields of selected variables**

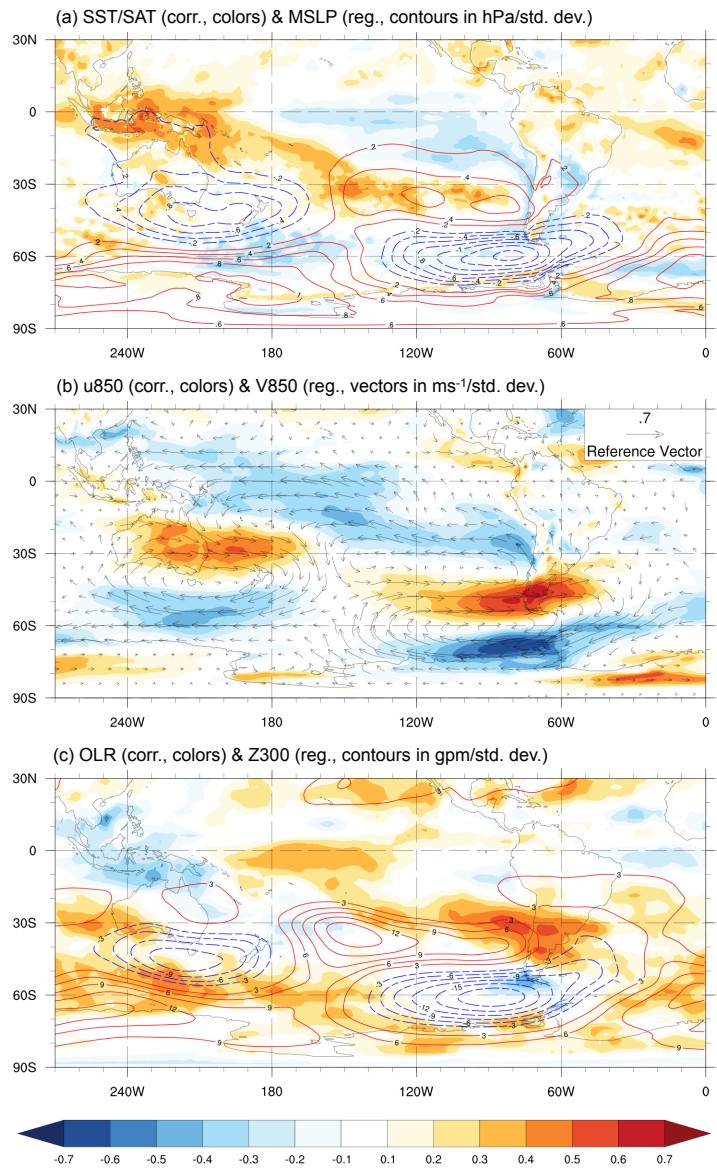

**Figure 9.** The same as in Fig. 8 but for the winter (April to September).

Low correlations are found between the main modes of interannual variability affecting the Patagonian climate and the
modeled time series. Our results show an almost null correlation between the CP index and the SMB time series at all timescales
considered. A higher but still not statistically significant correlation is found between the EP index and the SMB time series at
annual timescale ($r = -0.33$) as well as winter ($r = -0.15$) and summer ($r = -0.15$) timescales. Regarding the SAM index, a
very weak correlation was found between this index and the SMB time series at all timescales (see SAM pattern in Fig. S12).

**Annual SMB projected onto summer fields of selected variables**

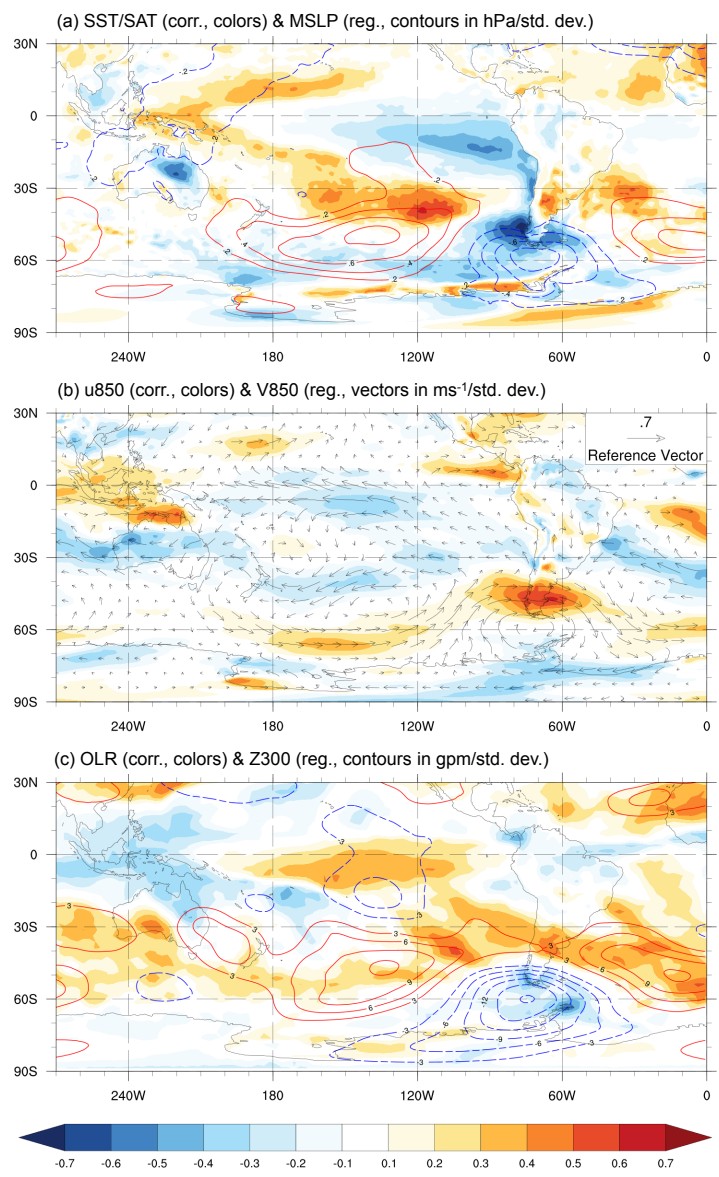

**Figure 10.** The same as in Fig. 8 but for the summer (October to March).

High correlation values are found between the Z300 Drake index and the modeled time series. For instance, there is a strong
and statistically significant correlation between Z300 Drake and the SMB time series at annual ($r = -0.65^*$), winter ($r = -0.66^*$), and summer ($r = -0.54^*$) timescales. Furthermore, the Z300 Drake index is highly correlated with the precipitation time series at all timescales, while there is a statistically significant correlation with the temperature time series only in summer ($r = 0.42^*$). Additionally, the T850 Drake index shows a strong correlation with the SMB time series at all timescales and a

**Table 2.** Correlation between indices of the main modes of interannual variability influencing Patagonia (and other custom indices) and the time series of spatially average fields of SMB, accumulated precipitation and mean temperature. (*) Statistically significant value at a significance level of 5%.

| | Annual correlation | | | Winter correlation | | | Summer correlation | | |
|---|---|---|---|---|---|---|---|---|---|
| | SMB | PRECIP | TEMP | SMB | PRECIP | TEMP | SMB | PRECIP | TEMP |
| Nino 3.4 | $-0.20$ | $-0.32$ | $-0.08$ | $-0.10$ | $-0.07$ | $-0.08$ | $-0.09$ | $-0.38^*$ | $-0.12$ |
| Nino 1+2 | $-0.27$ | $-0.25$ | $0.04$ | $-0.12$ | $0.00$ | $0.10$ | $-0.14$ | $-0.37^*$ | $-0.03$ |
| EP ENSO | $-0.33$ | $-0.24$ | $0.04$ | $-0.15$ | $-0.02$ | $0.12$ | $-0.15$ | $-0.20$ | $0.07$ |
| CP ENSO | $-0.04$ | $-0.16$ | $-0.05$ | $0.02$ | $-0.06$ | $-0.16$ | $-0.03$ | $-0.20$ | $-0.05$ |
| SAM | $-0.08$ | $0.10$ | $0.26$ | $-0.12$ | $-0.01$ | $0.18$ | $-0.12$ | $0.01$ | $0.07$ |
| Z300 Drake | $-0.65^*$ | $-0.60^*$ | $0.07$ | $-0.66^*$ | $-0.65^*$ | $-0.15$ | $-0.54^*$ | $-0.48^*$ | $0.42^*$ |
| T850 Drake | $-0.67^*$ | $-0.38^*$ | $0.46^*$ | $-0.46^*$ | $-0.29$ | $0.32$ | $-0.63^*$ | $-0.34$ | $0.63^*$ |
| SST-R2 | $-0.57^*$ | $-0.20$ | $0.41^*$ | $-0.24$ | $0.00$ | $0.34^*$ | $-0.66^*$ | $-0.25$ | $0.68^*$ |
| U850-R3 | $0.71^*$ | $0.86^*$ | $0.26$ | $0.85^*$ | $0.92^*$ | $0.52^*$ | $0.59^*$ | $0.80^*$ | $-0.15$ |

high correlation with the near-surface temperature time series during summer ($r = 0.63^*$). The highest correlation between the SMB time series and the set of indices explored is maintained with the U850-R3 index, at annual ($r = 0.71^*$), winter ($r = 0.85^*$), and summer ($r = 0.59^*$) timescales.

Finally, there is a strong coherence between the Z300 Drake index and the T850 Drake index at annual ($r = 0.79^*$), winter ($r = 0.76^*$), and summer ($r = 0.86^*$) timescales. Also, we found a statistically significant correlation between the SST-R2 index and the T850 Drake index at annual ($r = 0.49^*$) and summer ($r = 0.76^*$) timescales, and a strong coherence between the Z300 Drake index and the U850-R3 index at all timescales ($|r| > 0.73^*$).

### 3.7 Separate analysis for the NPI and the SPI

In order to ensure that our main results characterize the large-scale climatic control of the SMB of the NPI and the SPI separately, we computed the NPI-only and SPI-only annual SMB time series and then we repeated the analysis from Sect. 3.5 but for each series independently.

The NPI-only and SPI-only annual SMB time series show high correlations with the original SMB time series ($r = 0.81^*$ and $r = 0.99^*$, respectively). In turn, the correlation between the NPI-only and the SPI-only annual SMB time series is $r = 0.71^*$. The large-scale maps resulting from this separate analysis (Figs. S8 and S9) confirm the results obtained from the previous sections. In other words, years of relatively high (low) SMB are characterized by the presence of a MSLP center located around the Drake Passage that leads to anomalous circulation and the strengthening (weakening) of the zonal winds. Also, regional SST cooling (warming) is observed.

The separate analysis also shows some differences between the large-scale maps of each icefield. In the NPI-only case, the equatorial SST cooling is almost absent as well as the intensification of the trade winds. Additionally, the positive anomalies of MSLP to the north of the Drake low disappear, as well as the pattern of low, high and low anomalies of geopotential height in the upper troposphere that appeared to spread from the tropics to the extratropics. Moreover, the Drake low reduces its size and a high-pressure center establishes over the Amundsen sea. These features are specific of the NPI-only case, and reveal a more inhibited tropical signal in the climatic control of the SMB of the NPI when compared to the SPI. The SPI-only maps (Fig. S9) are virtually identical to the maps constructed using both icefields (Fig. 8). Thus, although our main results of Sect. 3.5 characterize the large-scale climatic control of the SMB of the NPI and the SPI separately, the atmospheric and surface oceanic connection with the tropics remains as a feature only of the SPI.

## 4 Discussion

Several sources of uncertainty are present in this study. To a large extent, these can be grouped into model-related uncertainties (e.g., calibration parameters, albedo parametrization, temperature thresholds, and downscaling methods) and uncertainties related to the meteorological input (e.g., mean values and interannual variability). In this investigation, we analyzed how the main sources of uncertainties could affect the representation of the annual SMB time series. We focused on the variability of the modeled SMB instead of its mean value since we explicitly calibrated the parameters of the SMB model to obtain similar values of mean annual SMB for the NPI and the SPI to those found in Minowa et al. (2021).

We handled uncertainties related to the modeling and mean value of the meteorological input by conducting several sensitivity experiments. Results showed that these sources of uncertainty do not affect the validity of the SMB time series estimate (see Sect. 3.2). In addition, the validity of the interannual variability of precipitation and near-surface temperature obtained from RegCMv4 was assessed by verifying against the CR2MET product, which reproduces with high accuracy the interannual variability of long-term weather stations in Patagonia (Figs. S2-S4). Results showed that precipitation and near-surface temperature from RegCMv4 have a good agreement with CR2MET in terms of interannual variability (see Fig. 2). Regarding insolation, the scarcity of in-situ measurements during 1980-2015 prevented us from verifying the interannual variability of that modeled field. Nonetheless, we base the validity of the interannual variability of insolation on the internal coherence of the model. We argue that since the model can well represent the variabilities of precipitation and near-surface temperature in the region, it necessarily has to reproduce the cloud cover variability adequately and, therefore, the variability of insolation.

All of the above considerations argue in favor of the robustness of our results. Moreover, a comparison with other studies of the SMB in Patagonia further supports our findings. For instance, in terms of interannual variability, our results are in good agreement with those from Schaefer et al. (2013, 2015). For the common period 1980-2010, the correlation between the annual SMB time series modeled in this study and Schaefer et al. (2013, 2015) reaches $r = 0.70^*$ for the NPI and $r = 0.73^*$ for the SPI. Also, years of strong SMB anomalies from Lenaerts et al. (2014) are consistent with our annual SMB time series (e.g., large negative anomalies in 1982, 1984, and 1987, and large positive anomalies in 1990, 2009, and 2010). To the south of the SPI, Möller and Schneider (2008) presented a modeled SMB time series for Gran Campo Nevado ice cap (52º50' S, 73º10'

W). Their SMB times series shows strong negative anomalies in 1982 and 1984 and large positive anomalies in 1990 and 1995,
consistent with our results. Further south (e.g., Tierra del Fuego), other SMB pattern seems to prevail (e.g., Buttstädt et al.,
2009), suggesting a southward limitation of the regional pattern.

The sensitivity experiment designed to test the dependence of the modeled SMB on the interannual variability of each
meteorological input allowed us to rule out solar radiation as a possible controlling variable of the SMB. Also, the experiment
showed that precipitation exerts the primary control over the SMB, followed by temperature (Table S9). This outcome is
consistent with the regression analysis results at the local scale (Table 1) by which annual anomalies in the SMB are related
primarily to anomalies in accumulation (highly correlated to precipitation) and secondarily to anomalies in ablation (highly
correlated to temperature). Similar findings have been found before, e.g., at Grey and Tyndall Glacier (Weidemann et al.,
2018b). Our results show that while accumulation anomalies are expected to dominate during winter and summer, ablation
anomalies are significant only during summer. Accordingly, the regression analysis at the local scale shows that years of
relatively high SMB show an increase in annual precipitation (greater in winter) and a decrease in summer temperatures,
while years of relatively low SMB are related to the opposite conditions. We argue that these as the most favorable local
meteorological conditions to produce anomalies in the SMB signal.

It is important to clarify that insolation is not a controlling variable of the SMB, even though it shows the second-highest
correlation with annual SMB ($r = -0.44^*$). For example, to compute the effect of an insolation anomaly on the surface energy
flux over a snow surface (see Eq. 5), we have to multiply the anomaly by a factor of 0.15. In contrast, in the case of temperature,
that factor increases to 9.5. A temperature anomaly of one standard deviation (0.37 ℃ if we use the annual time series standard
deviation to estimate the magnitude of typical deviations) implies the addition of 3.5 $\text{Wm}^{-2}$ to the surface energy flux. In
comparison, an insolation anomaly of one standard deviation (4 $\text{Wm}^{-2}$) adds only 0.6 $\text{Wm}^{-2}$. Thus, temperature anoma-
lies have a greater influence on the ablation field than insolation anomalies (with an approximate ratio of 7:1), even though
temperature shows the lowest correlation with the SMB. Interestingly, this finding does not depend on the low coefficient of
variation of the annual insolation (std. dev./mean near 3%) but on the relation between the typical anomalies of near-surface
temperature and insolation. On the other hand, we interpret the correlation between insolation and SMB as a by-product of the
correlation between insolation and precipitation ($r = -0.53^*$), which is due to the presence of clouds and the diminishing of
solar radiation associated with precipitation. As precipitation shows the best correlation with SMB among all meteorological
variables ($r = 0.69^*$), part of this coherence necessarily will be transferred to the insolation.

As our principal finding, we found a strong connection between the SMB of the Patagonian Icefields and the MSLP near the
Drake Passage at interannual timescales. Based on our results, we propose that years of relatively high SMB are characterized
by the presence of the Drake low (Fig 8a) that induces an enhancement of the westerlies impinging the Patagonian Icefields.
This, in turn, increases the precipitation via orographic enhancement (Roe, 2005; Garreaud et al., 2013). The Drake low is
thermodynamically maintained by a core of cold air that concurrently cools the Patagonian Icefields and the Pacific Ocean
adjacent to Patagonia (Fig. 5d), especially in summer (Fig. 7d) when this core is cooler and further north than that in winter
(Fig. 6d). We hypothesize that during winter, even if no cooling occurs, the core of cold air prevents the warming associated
with increased zonal winds in central Patagonia that would otherwise be expected under normal conditions (Garreaud et al.,

2013, see Table 2). In addition, during winter, the meridional gradient of geopotential height (Z300, Fig. 9c) to the west of Patagonia tends to be stronger than that during summer (Fig. 10c) due to a more pronounced high-pressure anomaly established over the subtropical Andes. This produces a greater increase in westerly winds and precipitation during winter than in summer (Figs. 6, 7). In this way, both the dynamics and thermodynamics associated with the Drake low would explain the increase in annual precipitation (greater in winter) and the decrease in summer temperature associated with years of relatively high SMB.

We found only weak correlations between the SMB and atmospheric modes of variability, such as the El Niño-Southern Oscillation (ENSO) and the Southern Annular Mode (SAM), implying little dependency between these modes and the SMB of the Patagonian Icefields (Table 2). We highlight that this result characterizes the present-day long-term (1980-2015) linear relationship between the annual variability of atmospheric modes and the SMB. One single event may profoundly impact the mass balance of the Patagonian Icefields (see for example Gómez et al. (2022)), which agrees with our results as long as the long-term linear relationship remains weak. Our results suggest that the low (high) pressure anomalies located over the Amundsen-Bellingshausen Sea during Central Pacific La Niña (El Niño) events (see Yuan et al. (2018) and references therein; also see our Fig. S11) are ineffective to enhance (reduce) the westerlies impinging the Patagonian Icefields due to westward displacement of the anomalous pressure center from the Drake Passage. Meanwhile, even though Eastern Pacific La Niña (El Niño) events are associated with the presence of low (high) pressure center anomalies near the Drake passage (Fig. S10), these anomalies appear to be much stronger during years of relatively high SMB than during years of relatively low EP index. Anyhow, the specific reasons why Eastern Pacific La Niña events are ineffective to produce years of relatively high SMB require further investigation. On the other hand, positive (negative) SAM phases exhibit slight strengthening (weakening) of the westerlies upstream of Patagonian Icefields and simultaneous anomalous warming (cooling) of Patagonia (Fig. S12). This means that for the SAM index the two observed processes tend to cancel each other out in developing SMB anomalies.

This study does not assess the teleconnections that potentially trigger the Drake low. However, we speculate that the origin of this pressure feature might be associated with tropical forcing due to the decreased convective activity over the central equatorial Pacific and increased convective activity over the western equatorial Pacific (e.g., Hoskins and Karoly, 1981; Karoly, 1989) observed during years of relatively high SMB (Fig 8c). As the SMB correlates better to the Eastern Pacific ENSO index than the Central Pacific ENSO index (Table 2), we argue that the establishment of the Drake low would be highly sensitive to the specific location of SST anomalies in the tropical Pacific. The low correlation between the Eastern Pacific ENSO index and the SMB could be a consequence of similar considerations in the sense that only certain eastern Pacific SST warming and cooling events could activate an anomalous pressure center near the Drake Passage. Additionally, we conjecture that the summer cooling of the Patagonian Icefields during years of relatively high SMB is mainly associated with the thermodynamics of the Drake low, as exposed previously, and not with the eastern Pacific SST cooling. This seems reasonable since nearly 40% of the variance of the summer temperature over the Patagonian icefields is explained by the lower tropospheric temperature near the Drake Passage (Table 2).

## 5 Conclusions

In this study, we investigated the present-day climatic control of the SMB of the Patagonian Icefields at interannual timescales. To do this, we first modeled the main surface meteorological and glaciological conditions during the period 1980-2015 and obtained a robust estimate of the annual anomalies of the spatially averaged SMB. Then, we used the time series of the SMB anomalies to derive regression and correlation maps against fields of climate variables, and to assess its relation with main atmospheric modes of variability at the interannual timescale. In this way, we determine the local, regional and large-scale climate processes controlling the annual SMB variations of the Patagonian Icefields. Our main findings are as follows:

– The interannual SMB variability of the Patagonian Icefields is controlled by the precipitation and near-surface temperature variabilities. Year-to-year SMB variations show almost no dependence on downward surface solar radiation variations.

– Regarding the local-scale conditions, years of higher than average SMB feature a higher than average annual accumulation and a lower than average summer ablation. Consistently, an increase of annual precipitation and a decrease of summer near-surface temperature are observed over the Patagonian Icefields. Opposite conditions are evident during the years with lower than average SMB.

– In relation to the regional-scale conditions, positive anomalies of the annual SMB are associated with an intensification of the westerly winds impinging the Patagonian Icefields and an increase of the precipitation in western Patagonia accompanied by drier conditions to the east of the Andes ridge. A regional decrease in near-surface temperatures is observed during summer, while null or little temperature changes are evident during winter. Negative anomalies of the annual SMB are associated with the opposite conditions.

– Concerning the large-scale conditions, years of relatively high SMB are characterized by the establishment of an anomalous low-pressure center near the Drake Passage, the Drake low, that induces an anomalous cyclonic circulation accompanied by enhanced westerlies impinging the Patagonian Icefields. The Drake low is thermodynamically maintained by a core of cold air that cools the Patagonian Icefields during summer. Years with lower than average SMB are associated with the opposite conditions.

– We found little dependency between the interannual SMB variability of the Patagonian Icefields and main atmospheric modes of variabilities such as SAM and warm and cold ENSO phases. Further work is required to understand the low annual correlation between EP ENSO index and the SMB of the Patagonian Icefields.

This research study gives new insights for understanding the complex interplay between the present-day climate processes and local-scale cryospheric processes in the southern Andean Cordillera. Low dependence of the Patagonian Icefields' SMB on main atmospheric modes of variability suggests a poor ability of ENSO and SAM indexes to reproduce the past and future interannual variability of the SMB. Instead, this study highlights the Drake Passage as a key region capable of reproducing the interannual variability of the SMB since it explains the linkage between large-scale processes and the SMB behavior

reasonably. Finally, findings from local-scale assessment facilitate the diagnostic of SMB anomalies in terms of precipitation, near-surface air temperature, and surface downward solar radiation anomalies, providing a conceptual framework useful for
future research in the area.

*Author contributions.*  TC conducted the investigation with the supervision of MR and RG. All co-authors contributed to the conceptualization of the research goals. TC implemented the model code, performed the simulations and prepared the manuscript with contributions from all co-authors.

*Competing interests.*  The authors declare that they have no conflict of interest.

*Acknowledgements.*  This research emerged from the leading author's master thesis project funded by the Chilean National Agency for Research and Development (CONICYT-PCHA/MagísterNacional/2016-22160660). We also thank to the Center for Climate and Resilience Research (CR2, CONICYT/FONDAP/15110009) for supporting with data and equipment to carry out this work, and for providing partial funding.

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
