# Peer review of "Climatic control of the surface mass balance of the Patagonian Icefields"

_EGUsphere, 2022_

## Author Comment (AC1)

Response to anonymous referee #1

We thank the referee for their constructive comments, which will certainly help to shape this manuscript into an improved paper. We provide here our responses to each comment and/or question made and how we will modify the manuscript as a result. The referee's comments are in blue, and our response in normal text.

Review of "Climatic control of the surface mass balance of the Patagonian Icefields" by Carrasco-Escaff et al., submitted to The Cryosphere.

The authors present an interesting study that adds valuable new knowledge to climate and glacier science related to southern South America. The study has been carried out well and is sufficiently documented over large parts of the manuscript. Just the description of the sensitivity analysis is in parts hard to follow and some efforts should be undertaken to improve readability of this section. Apart from this, I have two major objections that prevent me from supporting publication of the article in its present form:

Major comment 1)

The downscaling of solar radiation as it is described in the one sentence provided in L183f has to be questioned. Bilinear interpolation of shortwave radiation on a non-systematically varying surface (like a DEM representing natural terrain) leads to wrong values at the higher-resolution scale. The angle between incoming direct solar radiation and surface slope/aspect (incidence angle) is crucial in determining the right amount of energy reaching the glacier surface. Hence, simply interpolating radiation values from low- to high-resolution grids introduces errors that could easily double or halve solar radiation energy reaching the surface. Regarding diffuse radiation, the skyview factors of the high-resolution grid cells might probably differ considerably from those of low-resolution fields. Taken together, it requires more to downscale solar radiation than just bilinear interpolation.

As spatiotemporal variability of solar geometry can easily be implemented in a downscaling model, the approach needs to be refined by considering incidence angles at each grid cell of the high-resolution topography. Otherwise, the resulting values are simply wrong. Moreover, a validation needs to presented that compares original and downscaled values to in situ measurements (ideally at an on-glacier weather stations). Such a validation must also be presented for T and P, as otherwise it is hard to argue why the RegCM fields can be used for reliable SMB modeling, especially as they show considerable biases to the reference CR2MET climate, which are corrected in a rather simple way only. I'm sure that the team of authors has access to such data even if it might cover only a short period of time.

These validations might also help to overcome the problem of validating the modeled SMB with respect to inter- and intra-annual variability. Assuming that downscaled T, P and R clearly show seasonal variability on a local scale, this would also suggest that the modeled SMB might be reliable in this respect.

We thank the referee for this valuable comment. We performed bilinear interpolation on the solar radiation field motivated by reproducing the temporal (year-to-year, winter-to-winter, and summer-to-summer) variability of the SMB. The slope and aspect, understood as fixed features of the terrain, would have a stronger importance in the assessing of the spatial variability of the SMB than in its temporal variability. Nonetheless, the referee's warning

about the possible introduction of errors that could double or halve solar radiation needs to be addressed. In this regard, we will estimate the error introduced in the solar radiation by the use of bilinear interpolation in comparison with a downscaling technique considering incidence angles at each grid cell of the high-resolution topography. Then, we will compare the interannual variability of the modeled SMB using both techniques, and after that we will diagnose the need for changing the downscaling method for solar radiation. If there is a need to change the downscaling method for solar radiation to one considering incidence angles at each grid cell of the high-resolution topography, we will implement that change throughout the manuscript. If not, we will include the mentioned analysis in the supplementary material of a revised version of the manuscript and maintain the original bilinear interpolation technique.

Due to the nature of this study, we are interested in validating the meteorological variables on an interannual scale, which necessarily entails having large data periods. Other time scales than the one mentioned are beyond the scope of the investigation. As shown in fig. S1, there are few stations (and with few data) on or near the icefields to perform this validation. Bozkurt et al. (2019) perform a validation with these stations for the modeled variables of near surface temperature and precipitation. Furthermore, in this work we decided to compare the interannual variability of the RegCMv4 modeled variables with the CR2MET product, which combines the few observations in the area with reanalysis data. This comparison also allows us to have an idea of the possible bias of the modeled variables, which is useful to establish the intervals of possible bias in the sensitivity analysis. Given that, as mentioned, there is not enough data from meteorological stations to validate the interannual variability of the variables, the sensitivity analysis is the instrument with which we validate that the conclusions of our work do not vary substantially even in the presence of bias. Similarly, and despite the above, a revised version of the manuscript will compare the variables modeled with observational data for the stations that do have sufficient data.

Major comment 2)

Climate forcing is analyzed using the SMB integrated over NPI and SPI together. This spatially undifferentiated way of looking at the outcome of this study is a missed opportunity that should be accounted for in a revised and extended version of the study. In its present form the analysis prohibits to get an idea about potential regional variability of forcing mechanisms across Patagonia. I would like to see similar figures to Figs. 6-11 be added to the supplement that show the correlations with only NPI and SPI. Analyzing the differences of these two sets of maps/graphs would give valuable insight into regional variations of climate forcing across Patagonia. This would strengthen the interpretation of the so far presented results which just integrate over NPI and SPI. Sections 3.3-3.5, as well as discussion and conclusion should then be extended accordingly. As we know from the literate that NPI and SPI do not always show the same patterns of glacier change, such an analysis might be of really high value to science – even if it shows that climate forcing mechanisms do not differ significantly for NPI and SPI.

We thank the referee for this suggestion. We agree that performing the same analysis on NPI and SPI separately might be of really high value to science. Accordingly, we will develop figures similar to Figs. 6-11 for the NPI and the SPI, we will include them in the supplementary material, and we will discuss them in the manuscript.

In addition to these comments I have quite some minor comments that also needs some attention of the authors. Based on the two major comments above and the minor comments below, I suggest to return the manuscript for major revision.

Minor comments:

L9: better: ...fields of climate variables from the ERA-Interim…

We thank the referee for this suggestion. We will change this line of the manuscript accordingly.

L40: These positive trends fit to the recent southward shift and strengthening of the southern hemispheric westerly wind belt (e.g. Goyal et al. 2021, doi:10.1029/2020GL090849), which might be of interest here.

We thank the referee for this comment. We will refer to the strengthening and southward shift of the southern westerly wind belt when mentioning the observed precipitation trends in Patagonia.

L55-57: These moister than average conditions in southern Patagonia have already been suggested to significantly influence SMB (Möller et al. 2007, doi:10.3189/172756407782871530), which should be noted here.

We thank the referee for pointing this out. We will notice this in a revised version of the manuscript.

L80: better: …, i.e. the net change of mass at the surface, … "Gain" suggest an increasing mass of ice, but SMB has been positive and negative in the period studied. See Cogley et al. 2011 (Glossary of Glacier Mass Balance) for further details on the related terminology.

We thank the referee for this correction. We will change the line in the manuscript accordingly.

L81ff: I see no need to explain glacier mass balance in such detail as the manuscript is written for the cryosphere-centered journal. E.g. basal melting should only be mentioned if it is of interest at the glaciers modeled in the presented study.

We thank the referee for this recommendation. We will remove the detailed explanation of glacier mass balance from the manuscript.

L95ff: Braun et al. 2019 and Dussaillant et al. 2019 (both in the manuscript) should also be mentioned here. And it should be discussed that these two remote sensing studies have shown strong mass loss especially over the SPI, which contrasts the positive SMB mentioned before. In its present form the reader gets a picture of increasing ice masses in southern Patagonia, which is wrong.

We thank the referee for warning us about this possible misunderstanding. We will mention the references in the paragraph and we will emphasize the contrast between negative total mass balance quantified by remote sensing methods and the positive SMB obtained through modeling.

L129: Why ERA-Interim and not ERA5 which is available for quite a while now?

RegCMv4 simulations use initial and boundary conditions from ERA-Interim reanalysis because it was fully available at the time the simulations were designed and executed (year 2015), meanwhile ERA5 was not.

L134: Also provide reference to Alvarez-Garreton et al. 2018 here, and not only at the end of the paragraph.

We thank the referee for this recommendation. We will include a reference to Alvarez-Garreton et al. 2018 in L134.

L132-140: What makes the CR2MET dataset a reliable reference? I do not question here that it could be used as this, but I would greatly appreciate additional argumentation. It is necessary to outline and explain how well this dataset represents in situ conditions. Moreover, information about shortcomings and especially inaccuracies of the dataset are needed to be able to judge about its reliability. And finally (maybe most important) why are the RegCM fields created and used when CR2MET already exists? What is the advantage of RegCM over CR2MET and does this advantage justify the introduction of additional uncertainty (by comparing it to CR2MET before usage)?

Please see our response to major comment #1 in relation to CR2MET. We used the RegCMv4 simulations basically because, among the modeled variables, they include near surface temperature, precipitation and surface downward solar radiation, useful for the SMB modeling. Also, they come from a physical downscaling and therefore are physically coherent. Instead, the CR2MET uses statistical downscaling and does not have solar radiation among its output variables. We used the CR2MET product to estimate reasonable intervals for possible biases in near surface temperature and precipitation for the sensitivity analysis, especially in an area where there are only few stations with data between 1980-2015.

L147: better: "… of world-wide glacier extent at the beginning…", as "extension" implies a process of increase rather than a static condition

We thank the referee for pointing this out. We will modify the line accordingly.

L158: not clear what is meant here: "Lastly, we spatially unweighted averaged the meteorological forcing…"

We thank the referee for warning us about this unclear expression. In this step we computed the spatial average of the meteorological forcing assigning the same weight to each grid point. We will clarify this in a revised version of the manuscript.

L159: better: "Only grid points within…" (omit "Note that")

We thank the referee for this suggestion. We will change this line in the manuscript accordingly.

L192: provide reference for this representation of the fraction of solid precipitation

We thank the referee for this suggestion. We will provide reference for this representation in a revised version of the manuscript.

L209ff: It would be interesting to get some values on the distribution of snow/firn after the spin-up time: Give average numbers for snow-/firnline altitudes across the study area and discuss potential spatial variations in case they exist. Give reference to other studies which derived snowline altitudes in Patagonia and shortly compare your results to these findings.

We thank the referee for this valuable suggestion. Nonetheless, we think that an extension of our work analyzing the specific spatial distribution of the snow-/firnline altitude is beyond the scope of our work.

L231-235: This is a really nice idea. However, I strongly request that also information about the bias in SMB compared to the reference SMB is somehow incorporated in the Taylor diagram (e.g. by scaled sizes or color-scales of the points shown). The so far given information about correlation and standard deviation only give insight into how well the variability is represented, but do not tell anything about resulting biases.

We thank the referee for this suggestion. We agree that showing the information about the resulting biases would be valuable. We will incorporate the requested information in the Taylor diagram in a revised version of the manuscript.

L239-249: This is an interesting approach, but more information is needed here. First, give reference to studies that introduced or at least support your idea. Second, give more details on how you determined the variability in the dataset and how you subsequently removed it. Also here, a quantification of biases is needed in addition to the measures of variability.

We thank the referee for these recommendations. We will give more information about this step in a revised version of the manuscript. We will give references to studies that support our idea, give more details on the methods we used, and incorporate a table with the resulting biases in the supplementary material.

Fig. 5: I suggest to add a thin black line representing a zero SMB in the upper panel of the figure. This would increase readability and make positive and negative SMB years more easily distinguishable.

We thank the referee for this suggestion. We will add the suggested line in the upper panel of Fig. 5.

Table 2: Add information about the period represented by the given numbers to the caption.

We thank the referee for this suggestion. We will add the information in the caption.

L287: The fact that annual insolation shows a higher correlation to SMB than annual temperature further supports my initial request regarding a refined handling of solar radiation during downscaling.

Please see our reply to major comment #1.

L304ff: Isn't that a necessary result of the over-simplified radiation downscaling that has been applied? I mean, how can a local-scale control over the SMB can be present when the applied downscaling is not able to produce the requited local-scale variability? (see my initial major comment) This analysis/interpretation must be redone after the radiation downscaling has been improved.

Please see our reply to major comment #1.

L307-318: It now entirely clear what was done here. A linear regression results in intercept and slope of a regression line, which are both important for interpretation. However, this full information is missing in Table 4 and has to be added. It must also be included in the following discussion.

We thank the referee for this suggestion. We will include the information about the intercepts in a separate table and discuss it in the text.

We did not consider solar radiation in the regression analysis because we had already concluded from the sensitivity experiments (described in Sect. 2.3.4 and tabulated in Table 1) that solar radiation exerts a negligible control over the year-to-year, winter-to-winter and summer-to-summer variations of the SMB. In the regression analysis, we are interested only in the variables that exert control over the SMB directly (near surface temperature and precipitation) or indirectly (the rest of climatic variables).

Figs. 6b/7b: I recommend not to use red/green colors for the isolines as these colors are hard to differentiate for a lot of color-blind people.

We thank the referee for this suggestion. We will change the color of the isolines accordingly.

L410ff: It would greatly strengthen the findings of the study if comparisons to other long-term SMB time series at other Patagonian glaciers would be given. E.g. Möller & Schneider 2008 (doi:10.3189/172756408784700626) present a modeled SMB time series for Gran Campo Nevado ice cap south of the SPI. This time series e.g. shows the same strongly positive anomalies of SMB in 1990 and 1995, which supports the presented findings for SPI by showing that they fit nicely into the picture presented by other studies. Further south (e.g. Tierra del Fuego) other SMB pattern prevail (e.g. Buttstädt et al. 2009, doi:10.5194/adgeo-22-117-2009), suggesting a southward limitation of the regional pattern.

We thank the referee for this recommendation. We will compare the findings of our study with other modeled SMB including the ones mentioned.

L418: Doesn't this contradict the results that you presented before (see my comments on L287 and L304ff)? This should be clarified either here and/or above.

This does not contradict our results. On the one hand, the fact that insolation shows the second-highest correlation with annual SMB (r=-0.44*) does not necessarily imply that annual variations of insolation exert an effective control over the annual variation in SMB. We interpret the correlation between insolation and SMB as a mere consequence of the correlation between insolation and precipitation (r=-0.53*) due to the presence of clouds and the diminishing of solar radiation when precipitating. As precipitation shows the best correlation with SMB (r=0.69*), this covariance necessarily will be reflected in the insolation.

On the other hand, the local-scale analysis shows a negligible dependence of the annual SMB on insolation. To give a more detailed interpretation of this result, please consider again the eq. 5 and assume snow as the type of surface. To compute the effect on the

surface energy flux of an insolation anomaly we have to multiply the anomaly by a factor 0.15, while in the case of temperature this factor increases to 9.5. For instance, a temperature anomaly of one standard deviation (0.37 ºC if we use the std. dev. of the annual time series to estimate the magnitude of typical deviations) implies an addition of 3.5 W/m$^2$ to the surface energy flux while an insolation anomaly of one standard deviation (4 W/m$^2$) adds only 0.6 W/m$^2$. Thus, temperature anomalies have a greater influence on the ablation field than insolation anomalies (with an approximate ratio of 7:1). This does not contradict the previous result about correlations because the local-scale analysis assesses the causes of the year-to-year variations of the SMB, while the correlation analysis does not assess causality.

L418-426: This paragraph would benefit from some references to either figures or tables.

We thank the referee for pointing this out. We will reference the proper tables and figures in a revised version of the manuscript.

L456ff: References to other studies dealing with this or comparable issues would support your speculation and should be added and discussed shortly.

We thank the referee for this suggestion. We will include references to other studies dealing with comparable issues in a revised version of the manuscript.

L474: This thought has not come to my mind until now: Is there any significant interannual variability in solar radiation? Or is it largely time-invariant? I'm asking because of the frequent presence of clouds in Patagonia. If there is no significant interannual variability, it would be a necessary consequence that SMB variations show almost not dependence on it. This needs to be analyzed (and outlined in the results section) before giving this broad statement, in order to potentially put it into the right context.

Please refer to our reply to the minor comment on L418. Although the low coefficient of variation of the annual insolation (std. dev./mean near 3%), the SMB variations show almost no dependence on the insolation due to the mathematical relation between the albedo, the c1 calibration parameter and the typical anomalies of near surface temperature and insolation. In a revised version of the manuscript, we will incorporate this analysis into the discussions and outline in the results section.

L490: "SBM" needs to be corrected to "SMB"

We thank the referee for pointing this out. We will correct this in the manuscript.

---

## Author Comment (AC2)

Response to anonymous referee #2

We thank the referee for their constructive comments, which will certainly help to shape this manuscript into an improved paper. We provide here our responses to each comment and/or question made and how we will modify the manuscript as a result. The referee's comments are in blue, and our response in normal text.

Summary: this paper describes the climatic controls on the surface mass balance (SMB) of the North and South Patagonian Icefields (NPI, SPI). This is achieved by estimating the annual and seasonal SMB with a simple snow, firn and ice accumulation and ablation model, subsequently regressing the SMB-anomalies time series to a suite of local, regional and climate indices. Results indicate that winter precipitation and summer temperature anomalies are the main drivers of SMB interannual variability. Also, the authors find that a pressure anomaly over the Drake's passage (the Drake low) is the dominant feature related to SMB departures, seemingly driving increased westerly winds and cooler conditions off the coast of Patagonia. No significant correlation was found between the SMB and major climate indices such as ENSO, which confirms previous work published in the area.

General comments: this is a well written paper, and is a nice contribution to the understanding of the NPI and SPI present-day behavior. The authors have taken preemptive actions to prevent the inevitable modeling uncertainties from affecting their conclusions, by focusing on correlations/anomalies only and by ensuring that potential biases in the meteorological forcings of the SMB model don't result in major changes in the year-to-year variability, measured through correlation and standard deviation of the time series. The organization of the manuscript is very intuitive and the use of English language is appropriate but for a few minor issues. Because the analysis rests so strongly on the simulated mass balance, the manuscript should devote a bit more space to discussing the calibration of the four main parameters of the model, namely the threshold at which precipitation falls as snow (here set as 2°C), and the ablation parameters (albedo, $c_0$ and $c_1$). The sensitivity of the model to these parameters should in turn influence the interplay between precipitation and temperature during the accumulation season, and the relative influence of radiation and temperature during the ablation season. It may be that the main conclusions don't change with respect to what is shown in the current version, but so far the paper seems to gloss over this topic in a manner too succinct.

We thank the referee for these valuable comments. We realize the importance of devoting more space to discussing the calibration of the main parameters of the model. In a revised version of the manuscript, we will perform sensitivity analyses for each one of the calibration parameters, varying their values within appropriate intervals and assessing the temporal variability of the SMB associated with each outcome. We will show the results in separate Taylor diagrams. These figures will be included in the supplementary material and they will be discussed in the manuscript.

Specific comments:

L132: It is not clear to me what the verification of RegCMv4 against CR2MET intends to achieve. There are clear biases shown in Fig2, which could result from several factors. Because you have threshold term in accumulation that depends on T, this bias in temperature could have compounded effects on the simulated SMB correlations. Do you do anything after verifying the two products against each other?

We compare the RegCMv4 against the CR2MET in order to verify the interannual variability of the main variables and to get an estimation of the possible biases in temperature and precipitation. With these estimates we performed a sensitivity analysis in order to ensure that the interannual variability of the SMB would not change even if there were biases in temperature and precipitation as large as those indicated in Fig. 2c. In this manner, the verification of the RegCMv4 against the CR2MET let us guarantee that the temporal variability of the SMB is maintained even after considering the compound effects of biases in the main meteorological variables.

L201: snow, firn and ice should not be called "soil". Please use something like "land cover".

We thank the referee for pointing this out. We will use "type of surface" instead of "soil" when referring to snow, firn and ice.

L210: in modeling parlance, "true" has a very specific meaning. Please revise.

We thank the referee for pointing this out. We will reformulate the paragraph accordingly.

L218: See general comments regarding the detail that is needed about the SMB model calibration process. Also, why do you compare the 2000-15 simulation with the 2000-19 Minowa estimates? Is it not possible to compare a common period?

Please see our response to the general comments. Regarding the periods of comparison, Minowa et al. estimates were informed only for the full 2000-19 period. Consequently, we are only capable of comparing our results against that period.

L229: These biases could also result from inadequacies in the CR2MET product. In particular, if it is station-based, previous research has shown that meteorological station data in Patagonia is unreliable, particularly precipitation.

In this part of the paper, we used the CR2MET in order to estimate the magnitude of the possible biases in temperature and precipitation. We agree with the referee in that these biases could result also from inadequacies in the CR2MET product, but in this step we were motivated in deriving optimal intervals for performing the sensitivity analysis rather than determining the origin of the biases.

L231: a similar analysis could be performed by perturbing some of the model parameters (see general comments).

Please see our reply to general comments.

L260: If I understand correctly, AAO was calculated only for the 1979-2000 period? But SMB is available until 2015? Maybe it'd be useful to have a summary table with all datasets used, indicating time window, time-step, and citation.

We used the AAO data computed for every day from 1980-01-01 to 2015-12-31. The daily value of the AAO can be calculated for any date, projecting the daily height anomalies at 700 hPa poleward 20º S onto a particular atmospheric pattern. To derive that pattern (the leading mode of EOF analysis of monthly mean 700 hPa height) it is considered only the data between 1979 and 2000. In a revised version of the manuscript, we will include a summary table with all datasets used as suggested.

L272: please reword to remind the reader that all these numerical quantities are estimates from your model. Also, the fact that annual SMB is positive means that for the ice fields to be in equilibrium (or decreasing in mass, as the literature suggests) then calving should account for the excess mass. Is that right? Also: there appears to be a slight increasing trend in the simulated SMB? Could you comment on this?

We thank the referee for this comment. 1. We will reformulate the paragraph as suggested. 2. Yes, calving accounts for the excess of mass. We will mention this in a revised version of the manuscript. 3. Precisely, there is a slight increasing trend in the simulated SMB, associated with a positive trend in the simulated precipitation. Nonetheless, in this investigation, we decided not to focus on the trend of the variables in order to maintain the purpose of the investigation delimited.

L298: How do you interpret the fact that although insolation shows the second-highest correlation with annual SMB (line 287), then the local-scale control indicates exactly the opposite? Maybe I'm missing something, but these two results seem inconsistent. Please clarify. Is this result sensitive to assumptions regarding the seasonal evolution of snow, firn and ice albedo? Nevertheless, it is expected that, unlike glaciers in mediterranean regions, solar radiation should have a minor role compared to temperature in Patagonia. High relative humidity and high very persistent cloud cover are coherent with this result.

On the one hand, the fact that insolation shows the second-highest correlation with annual SMB (r=-0.44*) does not necessarily imply that annual variations of insolation exert an effective control over the annual variation in SMB. We interpret the correlation between insolation and SMB as a mere consequence of the correlation between insolation and precipitation (r=-0.53*) due to the presence of clouds and the diminishing of solar radiation when precipitating. As precipitation shows the best correlation with SMB (r=0.69*), this covariance necessarily will be reflected in the insolation.

On the other hand, the local-scale analysis shows a negligible dependence of the annual SMB on insolation. To give a more detailed interpretation of this result, please consider again the eq. 5 and assume snow as the type of surface. To compute the effect on the surface energy flux of an insolation anomaly we have to multiply the anomaly by a factor 0.15, while in the case of temperature this factor increases to 9.5. For instance, a temperature anomaly of one standard deviation (0.37 ºC if we use the std. dev. of the annual time series to estimate the magnitude of typical deviations) implies an addition of 3.5 W/m$^2$ to the surface energy flux while an insolation anomaly of one standard deviation (4 W/m$^2$) adds only 0.6 W/m$^2$. Thus, temperature anomalies have a greater influence on the ablation field than insolation anomalies (with an approximate ratio of 7:1). This does not contradict the previous result about correlations because the local-scale analysis assesses the causes of the year-to-year variations of the SMB, while the correlation analysis does not assess causality.

L406: a small detail: probably "good" is not an appropriate adjective for describing correlation.

We thank the referee for pointing this out. We will change this word to a more appropriate adjective in a revised version of the manuscript.

L446: suggest replacing "maintains" with "remains".

We thank the referee for this suggestion. We will change this word accordingly.

---

## Author Comment (AC3)

Response to anonymous referee #3

We thank the referee for their constructive comments, which will certainly help to shape this manuscript into an improved paper. We provide here our responses to each comment and/or question made and how we will modify the manuscript as a result. The referee's comments are in blue, and our response in normal text.

General comments

The study analyzes the control of the present-day climate on the surface mass balance of the Patagonian Icefields. The main goals of the study are clearly formulated, and the study is well structured and written over largest parts. In the Discussion section, a stronger comparison with and discussion of results of other SMB studies in the region could strengthen the findings. Overall, the study adds valuable knowledge to the understanding of the interaction between climate and glacier mass balance in the southern Andes. I have one major comment which needs addressing and revision before publishing the article, together with several minor comments.

Major comment

The main limitation of this study is the fact that there is no spatial analysis for the correlation of SMB with large-scale indices and climate. It is possible that e.g. SAM does have an important impact on the SMB of the southern SPI, however, not on the whole study site. Averaging over such a large area can cause different signals in different regions to equal out. The study site does stretch over a large latitudinal band, and we know from literature that the climate and glaciology of NPI and SPI can show different characteristics and patterns. This is taken into account by calibrating the SMB model for both icefields individually, but then ignored throughout the rest of the paper.

Overall, I think by the spatial averaging a lot of valuable information is lost. I advise to conduct a spatiotemporal analysis instead of averaging over the SPI and NPI in order to gain information about the regional variability of climatic control on the SMB in Patagonia.

We thank the referee for this valuable comment. We will conduct a spatiotemporal analysis and discuss it in a revised version of the manuscript. Specifically, we will perform a principal component analysis (PCA) on the time-lat-lon field of SMB and we will seek for the main modes of variability of the SMB. If there is one leading mode of spatiotemporal variability (accounting for most of the joint variance), we will compare its PC with the time series we have been using so far (spatial average over the NPI and the SPI) in order to ensure that our method give us a time series representative of the variability of most of the icefields' area. If there is more than one leading mode of spatiotemporal variability, we will redo Figs. 6 and 9, and Table 5 considering each mode of variability and we will discuss the results in the manuscript.

Minor comments

L1: "Northern and Southern Patagonian Icefields" should be "Northern and Southern Patagonian Icefield"

We thank the referee for pointing this out. We will modify the line accordingly in a revised version of the manuscript.

L56 & 76 & 482: "dryer" should be "drier"

We thank the referee for this correction. We will fix this typo in a revised version of the manuscript.

L88: The word "scenario" is strongly associated with climate scenarios, I recommend to reformulate

We thank the referee for this recommendation. We will reformulate the sentence in a revised version of the manuscript.

L93: "assess" to "assesses"

We thank the referee for this correction. We will fix this typo in a revised version of the manuscript.

L112-123: The paragraph about the study site is a bit short in my opinion. I would include some brief information about major differences between the two icefields (e.g., SPI many marine terminating glaciers; do we have substantial climatic differences between the two icefields?). A reference to Fig. 2a would make sense here.

We thank the referee for this suggestion. In a revised version of the manuscript we will extend the section about the study area including the topics suggested by the referee.

L125-131: Which exact variables are taken form RecCMv4?

We took the near-surface air temperature, precipitation, and surface downward solar radiation fields from the RegCMv4. We will include this information explicitly in L126-127 in a revised version of the manuscript.

L133: Why are two different versions used for precipitation and temperature?

We used the last versions available of the CR2MET products of precipitation and temperature. As these products are independent, there is no problem in using different versions for each variable.

L132-140: Both datasets, RefCMv4 and CR2MET, are (at least partly) based on ERA-Interim. I miss a comparison with an independent dataset. What about weather station data, or an independent Reanalysis dataset?

As shown in Fig. S1, there are very few stations near the Patagonian Icefields with enough data available for deriving climate statistics to compare with the RegCMv4 within the period 1980-2015. The CR2MET product uses this information in the best possible way and constitutes our best station-based approach in the region. Nonetheless, we will include a comparison between these stations and the RegCMv4 in a revised version of the manuscript.

L150: You used the abbreviation ERA-Interim before. Introduce it at the first mentioning, please.

We thank the referee for pointing this out. We will introduce ERA-Interim at the first mention, as suggested.

L154: Dot is missing at the end of the sentence.

We thank the referee for this correction. We will fix this in a revised version of the manuscript.

L158: This is not clear to me: "Lastly, we spatially unweighted averaged the meteorological forcing and the glaciological over the Patagonian Icefields…"

We thank the referee for warning us about this unclear expression. In this step we computed the spatial average of the meteorological forcing assigning the same weight to each grid point. We will clarify this in a revised version of the manuscript.

L164: The first "DEM" can be removed.

We thank the referee for this suggestion. We will remove the first "DEM" in a revised version of the manuscript.

L183f.: This is not a downscaling of radiation, but simple interpolation.

We thank the referee for pointing this out. We will change the expression accordingly.

L199: I would replace the 10800s in the equation by a variable representing the timestep

We thank the referee for this suggestion. We will modify the equation as suggested in a revised version of the manuscript.

L201: These are not soil. Rather call it type of surface.

We thank the referee for pointing this out. We will use "type of surface" instead of "soil" when referring to snow, firn and ice.

L208: Accurately, the end of summer season would be the 31 March.

We thank the referee for pointing this out. We will refer to April 1st as the start of the autumn season.

L218: Please, use a consistent number of decimal places.

We thank the referee for pointing this out. We will use a consistent number of decimal places in a revised version of the paper.

L221-223: The values for c0 are very different between NPI and SPI. Why is this the case?

These values reflect the difference in SMB between NPI and SPI having similar sensitivity to near surface temperature.

L227: See comment to L132-140.

Please see our response to the minor comment on L132-140.

L241 & Table1: I recommend using a different abbreviation for the time period here to avoid confusion, as T has been used for temperature before.

We thank the referee for pointing this out. We will use a different abbreviation for the time period in a revised version of the manuscript.

Table 1 and following tables: It is common to put the table captions above the respective table.

We thank the referee for this recommendation. We will put the table captions above the respective table in a revised version of the manuscript.

Table 2: The annual SMB and precipitation value does not exactly add up from winter and summer values. Rounding error?

Yes, winter and summer value do not add exactly in the case of annual SMB and precipitation because of rounding errors.

L286-289: Only mention the significant correlations here: "Among the modeled meteorological variables, the annual SMB is found to have the largest correlation with the annual precipitation (r = 0.69), followed by annual insolation (r = −0.44) (see Table 3). The same order is also evident in winter. The correlation between the SMB and temperature is only significant in summer."

We thank the referee for this recommendation. We will modify the paragraph accordingly in a revised version of the manuscript.

L332: The correlation seems to be highest especially over the SPI?

While the map does not show correlation directly, effectively it shows the highest slope of regression over the SPI.

L346: "shows" to "show"

We thank the referee for this correction. We will fix it in a revised version of the manuscript.

Fig. 6b: The grey and white shading is confusing at the first glance, as it seems like there would be two different variables in this plot like it is in panel d. Maybe you can give the shading the same color as the contours to make it clearer.

We thank the referee for this suggestion. We will modify the figure and give the shading the same color as the contours as suggested.

L368f.: Refer to Fig. 9a here first.

We thank the referee for this suggestion. We will refer to Fig. 9a in L368 accordingly.

L392ff.: The low correlation with the ENSO and SAM could be due to the spatial averaging over the whole study site. Consider differentiating into regions.

Please see our reply to the major comment.

L418-426: Discussion and comparison with other SMB studies in southern Patagonia would support your findings. Similar findings have been found before, e.g., at Grey and Tynall Glacier (Weidemann 2018, https://doi.org/10.3389/feart.2018.00081)

We thank the referee for this suggestion. We will discuss and compare with other SMB studies as suggested.

We thank the referee for this correction. We will fix this typo in a revised version of the manuscript.

L476-489: Every paragraph starts with "years of … SMB are characterized by …". Consider reformulating.

We thank the referee for this suggestion. We will rephrase the paragraphs accordingly in a revised version of the manuscript.

---

## Author Response (AR1)

Santiago, Chile; December 29, 2022

Ref.: egusphere-2022-603

Dr. Tobias Sauter

Editor

The Cryosphere

Dear Dr. Tobias Sauter,

We are pleased to submit the revised version of our work "Climatic control of the surface mass balance of the Patagonian Icefields". It took us eight weeks to produce this new version because we performed further analyses along with the corrections and clarifications in the main text.

We appreciate the constructive and detailed comments provided by the three anonymous reviewers, which certainly helped to shape this manuscript into an improved paper. We are confident that the new version of our manuscript will be more interesting and accessible to the readers of The Cryosphere. Please find enclosed our detailed, point-by-point responses to the reviewers' comments. Here is a summary of the major changes to the manuscript that have been made to address the issues raised by the reviewers:

1.  Throughout the manuscript, we emphasized that we seek to obtain a robust estimation of the **interannual variability of the SMB**, which is the main motivation of our study, rather than getting exact estimates for the mean values of the modeled variables.

2.  We **improved the validation of the CR2MET dataset** (e.g., Sect. 2.2 and Figs. S2-S4). We compared annual precipitation and temperature estimates against available weather station measurements in Patagonia during 1980-2015. Even though CR2MET shows a slight cold and wet bias, it reproduces the interannual variability of the available time series with high accuracy. We also made it clearer why we used RegCMv4 instead of CR2MET and mentioned other studies that evaluated RegCMv4 in the region.

3.  We added several **sensitivity experiments** in order to guarantee the robustness of our results (e.g., Sect. 2.3.4, Sect. 3.2, and Tables S6-S9). We devoted a particular section of the manuscript to the motivation and design of these sensitivity experiments. The new experiments allowed us to determine the sensitivity of the modeled SMB to the main model parameters (calibration parameters, temperature thresholds and albedo parametrizations), the meteorological input (mean value and variability), and the complexity of the solar radiation remapping (see below).

4. We included a thorough evaluation of the uncertainties arising from the interpolation of **shortwave radiation** (e.g., Sect. 2.3.4, Sect. 3.2, and Table S9). We assessed our simple approach by comparing it with an alternative downscaling technique that considers terrain parameters such as slope and aspect. A detailed description of this alternative method is given in Sect. 2.3.4. We compared the modeled SMB using the original versus the alternative method, and tabulated the main results in Table S9. Our results show a decrease in the mean insolation and an increase in the mean SMB, but virtually no changes in the interannual variability of the SMB.

5. We conducted a **spatiotemporal analysis** on the SMB field using Empirical Orthogonal Function (EOF) analysis (e.g., Sect 3.1 and Figs. S6 and S7). We found that the annual time series of the spatially averaged SMB used in this work virtually coincides with the leading mode of the interannual variability of the SMB, which explains 59% of the total variance, dominating most of the area of the Patagonian Icefields.

6. We provide a **separate analysis for the NPI and the SPI** (e.g., Sect. 3.1 and Figs. S8 and S9). We computed the NPI-only and SPI-only annual SMB time series and repeated the analysis from Sect. 3.5 for each series independently. The separate analysis confirms our main results, yet, a more inhibited tropical signal in the climatic control of the NPI-only annual SMB is evident compared to the SPI-only annual SMB.

7. We **improved the discussion section** by adding paragraphs devoted to the handling of uncertainty, the comparison with other studies of SMB in Patagonia, and the role of insolation in the SMB. We also rephrased some of the conclusions following the advice of an anonymous reviewer.

We would like to acknowledge the reviewers again for their time and effort, and their insightful comments and suggestions.

We look forward to hearing from you.

Best regards,

Tomás Carrasco-Escaff, on behalf of all the authors.

The referee's comments are in blue, our response in normal text, and the specifics actions are highlighted in bold.

Review of "Climatic control of the surface mass balance of the Patagonian Icefields" by Carrasco-Escaff et al., submitted to The Cryosphere.

The authors present an interesting study that adds valuable new knowledge to climate and glacier science related to southern South America. The study has been carried out well and is sufficiently documented over large parts of the manuscript. Just the description of the sensitivity analysis is in parts hard to follow and some efforts should be undertaken to improve readability of this section. Apart from this, I have two major objections that prevent me from supporting publication of the article in its present form:

Major comment 1)

The downscaling of solar radiation as it is described in the one sentence provided in L183f has to be questioned. Bilinear interpolation of shortwave radiation on a non-systematically varying surface (like a DEM representing natural terrain) leads to wrong values at the higher-resolution scale. The angle between incoming direct solar radiation and surface slope/aspect (incidence angle) is crucial in determining the right amount of energy reaching the glacier surface. Hence, simply interpolating radiation values from low- to high-resolution grids introduces errors that could easily double or halve solar radiation energy reaching the surface. Regarding diffuse radiation, the skyview factors of the high-resolution grid cells might probably differ considerably from those of low-resolution fields. Taken together, it requires more to downscale solar radiation than just bilinear interpolation.

As spatiotemporal variability of solar geometry can easily be implemented in a downscaling model, the approach needs to be refined by considering incidence angles at each grid cell of the high-resolution topography. Otherwise, the resulting values are simply wrong. Moreover, a validation needs to presented that compares original and downscaled values to in situ measurements (ideally at an on-glacier weather stations). Such a validation must also be presented for T and P, as otherwise it is hard to argue why the RegCM fields can be used for reliable SMB modeling, especially as they show considerable biases to the reference CR2MET climate, which are corrected in a rather simple way only. I'm sure that the team of authors has access to such data even if it might cover only a short period of time.

These validations might also help to overcome the problem of validating the modeled SMB with respect to inter- and intra-annual variability. Assuming that downscaled T, P and R clearly show seasonal variability on a local scale, this would also suggest that the modeled SMB might be reliable in this respect.

R. We thank the referee for this valuable comment. In order to address this issue, **we implemented an alternative downscaling technique for the surface downward solar radiation that considers terrain parameters such as slope and aspect.** A detailed description of this alternative method is given in Sect. 2.3.4. Then, we compared the modeled SMB using our original, simple approach versus the alternative method. We tabulated the main statistics in Table S9. Our results show a decrease in the mean insolation and an increase in the mean SMB but virtually no changes in the interannual variability of the spatially averaged SMB.

Since we assessed the climatic control of the SMB based on the annual time series of the spatially averaged SMB anomalies, our main results are insensitive to the mean value of the SMB. Thus, even though by using our simple approach we could introduce errors in the solar radiation estimates, these errors do not alter the interannual variability of the spatially averaged SMB and, thus, do not alter our conclusions. **Since the conclusions of our study did not change with the method selected, we preferred to keep the simple approach and incorporate the information about the alternative**

**downscaling (description and results) in the manuscript.** Additionally, in order to further clarify this point, throughout this revised version of the manuscript, we emphasized that this study seeks to obtain a robust estimation of the interannual variability of the spatially averaged SMB rather than getting exact estimates of the mean values or the spatial variability of the modeled variables.

We could not have access to long records of in-situ measurements of solar radiation in the Patagonian Icefields' area to validate the interannual variability of the radiation directly against observations. Since this study focuses on the spatially averaged SMB at interannual timescales, short records measured over a few points could lead to wrong conclusions. Instead, to address this issue, **we performed a sensitivity analysis to determine our results' dependence on the mean value of the surface downward solar radiation**. A detailed description of this analysis is given in Sect. 2.3.4. The results showed that the interannual variability of the SMB was insensitive to reasonable changes in the mean value of the solar radiation.

Finally, **we clarified the usage of RegCMv4 and the CR2MET dataset** (please see L146-162). First, **we validated CR2MET against weather station measurements in Patagonia during 1980-2015**. We found that CR2MET has a cold and wet bias, but it reproduces the interannual variability of the available time series with high accuracy. Then, we used the CR2MET as the best estimate for temperature and precipitation available over the Patagonian Icefields' area and compared the RegCM4 against it. **The RegCMv4 model well-reproduces the interannual variability of the CR2MET temperature and precipitation**, but this is not the case for the mean values. As we found differences in the mean values of RegCMv4 versus CR2MET and considering that CR2MET has its own biases, we extended the sensitivity analysis of the SMB to the mean values of temperature and precipitation. Our results showed that the interannual variability of the SMB is insensitive to potential biases in temperature and precipitation (please see Table S7). **In this way, we found that RegCMv4 fields can effectively be used for reliable modeling of the interannual variability of the SMB**.

Major comment 2)

Climate forcing is analyzed using the SMB integrated over NPI and SPI together. This spatially undifferentiated way of looking at the outcome of this study is a missed opportunity that should be accounted for in a revised and extended version of the study. In its present form the analysis prohibits to get an idea about potential regional variability of forcing mechanisms across Patagonia. I would like to see similar figures to Figs. 6-11 be added to the supplement that show the correlations with only NPI and SPI. Analyzing the differences of these two sets of maps/graphs would give valuable insight into regional variations of climate forcing across Patagonia. This would strengthen the interpretation of the so far presented results which just integrate over NPI and SPI. Sections 3.3-3.5, as well as discussion and conclusion should then be extended accordingly. As we know from the literate that NPI and SPI do not always show the same patterns of glacier change, such an analysis might be of really high value to science – even if it shows that climate forcing mechanisms do not differ significantly for NPI and SPI.

R. We thank the referee for this suggestion. **In the revised version of the manuscript, we provide a separate analysis for the NPI and the SPI** (Sect. 3.1 and Figs. S8 and S9). In this analysis, we computed the NPI-only and SPI-only annual SMB time series and repeated the analysis from Sect. 3.5 (i.e., maps of large-scale control) for each series independently. **The separate analysis confirms our main results**, i.e., years of relatively high (low) SMB are characterized by the presence of a pressure center located around the Drake Passage that leads to anomalous circulation and the strengthening (weakening) of the zonal winds. Yet, a less marked tropical signal in the climatic control of the NPI-only annual SMB is evident compared to the SPI-only annual SMB.

Furthermore, in response to the major comment of referee #3, we conducted a spatiotemporal analysis on the SMB field using Empirical Orthogonal Function (EOF) analysis (Sect 3.1 and Figs. S6 and S7). We found that the annual time series of the spatially averaged SMB used in this work virtually coincides with the leading mode of the interannual variability of the SMB, which explains 59% of the total variance, dominating most of the area of the Patagonian Icefields (NPI and SPI both included).

Consequently, our annual SMB time series well-represents the variability of most of the Patagonian Icefields.

In addition to these comments I have quite some minor comments that also needs some attention of the authors. Based on the two major comments above and the minor comments below, I suggest to return the manuscript for major revision.

Minor comments:

L9: better: ...fields of climate variables from the ERA-Interim…

R. Advice taken. **We reworded the text as suggested in L9**.

L40: These positive trends fit to the recent southward shift and strengthening of the southern hemispheric westerly wind belt (e.g. Goyal et al. 2021, doi:10.1029/2020GL090849), which might be of interest here.

R. We thank the referee for this comment. **We mentioned the strengthening and southward shift of the southern westerly wind belt in L58-59.**

L55-57: These moister than average conditions in southern Patagonia have already been suggested to significantly influence SMB (Möller et al. 2007, doi:10.3189/172756407782871530), which should be noted here.

R. We thank the referee for pointing this out. **We noted this in L62-64**.

L80: better: …, i.e. the net change of mass at the surface, … "Gain" suggest an increasing mass of ice, but SMB has been positive and negative in the period studied. See Cogley et al. 2011 (Glossary of Glacier Mass Balance) for further details on the related terminology.

R. Advice taken. We corrected the text as suggested. **Please see L84**.

L81ff: I see no need to explain glacier mass balance in such detail as the manuscript is written for the cryosphere-centered journal. E.g. basal melting should only be mentioned if it is of interest at the glaciers modeled in the presented study.

R. We thank the referee for this recommendation. **We removed the detailed explanation of glacier mass balance from the manuscript**.

L95ff: Braun et al. 2019 and Dussaillant et al. 2019 (both in the manuscript) should also be mentioned here. And it should be discussed that these two remote sensing studies have shown strong mass loss especially over the SPI, which contrasts the positive SMB mentioned before. In its present form the reader gets a picture of increasing ice masses in southern Patagonia, which is wrong.

R. We thank the referee for warning us about this possible misunderstanding. We mentioned the references in the paragraph and emphasized the contrast between negative total mass balance quantified by remote sensing methods and the positive SMB obtained through modeling. **Please see L95-99**.

L129: Why ERA-Interim and not ERA5 which is available for quite a while now?

R. RegCMv4 simulations use initial and boundary conditions from ERA-Interim reanalysis because it was fully available at the time the simulations were designed and executed (year 2015), meanwhile ERA5 was not.

L134: Also provide reference to Alvarez-Garreton et al. 2018 here, and not only at the end of the paragraph.

R. We thank the referee for this recommendation. A reference to Alvarez-Garreton et al. 2018 **was included in L148**.

L132-140: What makes the CR2MET dataset a reliable reference? I do not question here that it could be used as this, but I would greatly appreciate additional argumentation. It is necessary to outline and explain how well this dataset represents in situ conditions. Moreover, information about shortcomings and especially inaccuracies of the dataset are needed to be able to judge about its reliability. And finally (maybe most important) why are the RegCM fields created and used when CR2MET already exists? What is the advantage of RegCM over CR2MET and does this advantage justify the introduction of additional uncertainty (by comparing it to CR2MET before usage)?

R. Please see our response to major comment #1.

L147: better: "… of world-wide glacier extent at the beginning…", as "extension" implies a process of increase rather than a static condition

R. We thank the referee for pointing this out. **We modified the line as suggested (please see L169)**.

L158: not clear what is meant here: "Lastly, we spatially unweighted averaged the meteorological forcing…"

R. We thank the referee for warning us about this unclear expression. In this step we computed the spatial average of the meteorological forcing assigning the same weight to each grid point. **We reworded this line in the revised version of the manuscript (please see L195-197).**

L159: better: "Only grid points within…" (omit "Note that")

R. Done.

L192: provide reference for this representation of the fraction of solid precipitation

R. We thank the referee for this suggestion. **We provided reference for this representation in L229-230**.

L209ff: It would be interesting to get some values on the distribution of snow/firn after the spin-up time: Give average numbers for snow-/firnline altitudes across the study area and discuss potential spatial variations in case they exist. Give reference to other studies which derived snowline altitudes in Patagonia and shortly compare your results to these findings.

R. We thank the referee for this valuable suggestion. Nonetheless, we think that an extension of our work analyzing the specific spatial distribution of the snow-/firnline altitude is beyond the scope of our work.

L231-235: This is a really nice idea. However, I strongly request that also information about the bias in SMB compared to the reference SMB is somehow incorporated in the Taylor diagram (e.g. by scaled sizes or color-scales of the points shown). The so far given information about correlation and standard deviation only give insight into how well the variability is represented, but do not tell anything about resulting biases.

R. We thank the referee for this suggestion. We agree that giving information about the resulting biases would be valuable. After thinking carefully, we decided to present the information as a table,

which include the data from the Taylor diagram and the information about the biases, among other statistics. **Please see Table S7**.

L239-249: This is an interesting approach, but more information is needed here. First, give reference to studies that introduced or at least support your idea. Second, give more details on how you determined the variability in the dataset and how you subsequently removed it. Also here, a quantification of biases is needed in addition to the measures of variability.

R. We thank the referee for these recommendations. We reformulated this section and incorporate as an additional sensitivity experiment in Sect. 2.3.4. We gave more information about this step in the revised version of the manuscript **(please see L313-319),** and incorporated a table with the resulting biases in the supplementary material (**please see Table S9**). We think there is no need for further references in the way the experiment is currently presented.

Fig. 5: I suggest to add a thin black line representing a zero SMB in the upper panel of the figure. This would increase readability and make positive and negative SMB years more easily distinguishable.

R. Done. **Please see Fig. 4**.

Table 2: Add information about the period represented by the given numbers to the caption.

R. Done. **Please see Table S1**.

L287: The fact that annual insolation shows a higher correlation to SMB than annual temperature further supports my initial request regarding a refined handling of solar radiation during downscaling.

R. Please see our reply to major comment #1.

L304ff: Isn't that a necessary result of the over-simplified radiation downscaling that has been applied? I mean, how can a local-scale control over the SMB can be present when the applied downscaling is not able to produce the requited local-scale variability? (see my initial major comment) This analysis/interpretation must be redone after the radiation downscaling has been improved.

R. Our results suggest that the referred result depends primarily on the interannual variability of the variables. When we use a downscaling method for solar radiation that considers terrain parameters such as slope and aspect, we obtain the same result as the one referred to in the comment. Please see our reply to the major comment #1.

L307-318: It now entirely clear what was done here. A linear regression results in intercept and slope of a regression line, which are both important for interpretation. However, this full information is missing in Table 4 and has to be added. It must also be included in the following discussion.

R. Since the independent variable has zero mean, the linear regression intercepts correspond to each dependent variable's mean values. Thus, we tabulated only the slope of each regression in order not to repeat the information showed in Table S1. **We mentioned this in L387-388**.

L325ff: Why is solar radiation not considered here?

R. We did not consider solar radiation in the regression analysis because we had already concluded from the sensitivity experiments (described in Sect. 2.3.4, Sect. 3.2 and tabulated in Table S9) that solar radiation exerts a negligible control over the year-to-year, winter-to-winter and summer-to-summer variations of the SMB. In the regression analysis, we are interested only in the variables that exert control over the SMB directly (near surface temperature and precipitation) or indirectly (the rest of climatic variables).

Figs. 6b/7b: I recommend not to use red/green colors for the isolines as these colors are hard to differentiate for a lot of color-blind people.

R. Advice taken. **We changed the color of the isolines as suggested (please see Figs. 5-7)**.

L410ff: It would greatly strengthen the findings of the study if comparisons to other long-term SMB time series at other Patagonian glaciers would be given. E.g. Möller & Schneider 2008 (doi:10.3189/172756408784700626) present a modeled SMB time series for Gran Campo Nevado ice cap south of the SPI. This time series e.g. shows the same strongly positive anomalies of SMB in 1990 and 1995, which supports the presented findings for SPI by showing that they fit nicely into the picture presented by other studies. Further south (e.g. Tierra del Fuego) other SMB pattern prevail (e.g. Buttstädt et al. 2009, doi:10.5194/adgeo-22-117-2009), suggesting a southward limitation of the regional pattern.

R. We thank the referee for this recommendation. **We compared the findings of our study with other modeled SMB as suggested in L524-532**.

L418: Doesn't this contradict the results that you presented before (see my comments on L287 and L304ff)? This should be clarified either here and/or above.

R. This does not contradict our results. **We devoted a paragraph in the discussion explaining this point (please see L545-557)**.

L418-426: This paragraph would benefit from some references to either figures or tables.

R. Advice taken.

L456ff: References to other studies dealing with this or comparable issues would support your speculation and should be added and discussed shortly.

R. We thank the referee for this suggestion. **We included references to other studies dealing with comparable issues in L522-531**.

L474: This thought has not come to my mind until now: Is there any significant interannual variability in solar radiation? Or is it largely time-invariant? I'm asking because of the frequent presence of clouds in Patagonia. If there is no significant interannual variability, it would be a necessary consequence that SMB variations show almost not dependence on it. This needs to be analyzed (and outlined in the results section) before giving this broad statement, in order to potentially put it into the right context.

R. **We address this point in L543-555**. Although the low coefficient of variation of the annual insolation (std. dev./mean near 3%), the SMB variations show almost no dependence on the insolation due to the relation between the typical anomalies of near surface temperature and insolation.

L490: "SBM" needs to be corrected to "SMB"

R. Done.

Response to anonymous referee #2

The referee's comments are in blue, our response in normal text, and the specifics actions are highlighted in bold.

Summary: this paper describes the climatic controls on the surface mass balance (SMB) of the North and South Patagonian Icefields (NPI, SPI). This is achieved by estimating the annual and seasonal SMB with a simple snow, firn and ice accumulation and ablation model, subsequently regressing the SMB-anomalies time series to a suite of local, regional and climate indices. Results indicate that winter precipitation and summer temperature anomalies are the main drivers of SMB interannual variability. Also, the authors find that a pressure anomaly over the Drake's passage (the Drake low) is the dominant feature related to SMB departures, seemingly driving increased westerly winds and cooler conditions off the coast of Patagonia. No significant correlation was found between the SMB and major climate indices such as ENSO, which confirms previous work published in the area.

General comments: this is a well written paper, and is a nice contribution to the understanding of the NPI and SPI present-day behavior. The authors have taken preemptive actions to prevent the inevitable modeling uncertainties from affecting their conclusions, by focusing on correlations/anomalies only and by ensuring that potential biases in the meteorological forcings of the SMB model don't result in major changes in the year-to-year variability, measured through correlation and standard deviation of the time series. The organization of the manuscript is very intuitive and the use of English language is appropriate but for a few minor issues. Because the analysis rests so strongly on the simulated mass balance, the manuscript should devote a bit more space to discussing the calibration of the four main parameters of the model, namely the threshold at which precipitation falls as snow (here set as 2°C), and the ablation parameters (albedo, $c_0$ and $c_1$). The sensitivity of the model to these parameters should in turn influence the interplay between precipitation and temperature during the accumulation season, and the relative influence of radiation and temperature during the ablation season. It may be that the main conclusions don't change with respect to what is shown in the current version, but so far the paper seems to gloss over this topic in a manner too succinct.

R. We thank the referee for these valuable comments. **In the revised version of the manuscript, we performed sensitivity analyses for each one of the main model parameters** (calibration parameters $c_0$ and $c_1$, temperature threshold and albedo), varying their values within appropriate intervals and assessing the temporal variability of the SMB associated with each outcome. Main statistics comparing outcomes with the original SMB series were tabulated in the supplementary material (please see Table S6). The sensitivity analyses are described in Sect. 2.3.4. and its results in Sect. 3.2. In general, results show that the interannual variability of the SMB was insensitive to the selected variations in the main model parameters, thus, our main conclusions did not change with respect to what is shown in the previous version of the manuscript.

Specific comments:

L132: It is not clear to me what the verification of RegCMv4 against CR2MET intends to achieve. There are clear biases shown in Fig2, which could result from several factors. Because you have threshold term in accumulation that depends on T, this bias in temperature could have compounded effects on the simulated SMB correlations. Do you do anything after verifying the two products against each other?

R. **We clarified the usage of RegCMv4 and the CR2MET dataset** (please see L146-162). We used the CR2MET as the best estimate for temperature and precipitation available over the Patagonian Icefields' area. To support this idea, **we verified CR2MET against weather station measurements in Patagonia** (but outside the Patagonian Icefields' area) during 1980-2015. We found that CR2MET has a cold and wet bias, but it reproduces the interannual variability of the available time series with high accuracy. Then, we compared the RegCM4 against CR2MET. **The RegCMv4 model well-reproduces the interannual variability of the CR2MET temperature and precipitation**, but this is

not the case for the mean values. As we found differences in the mean values of RegCMv4 versus CR2MET and considering that CR2MET has its own biases, after verifying the two products we performed a sensitivity analysis of the SMB to the mean values of temperature and precipitation. **In this revised version of the manuscript, we extended the previous analysis including more experiments**. Our results showed that the interannual variability of the SMB is insensitive to potential biases in temperature and precipitation (please see Table S7). **In this way, we found that RegCMv4 fields can effectively be used for reliable modeling of the interannual variability of the SMB**.

L201: snow, firn and ice should not be called "soil". Please use something like "land cover".

R. We thank the referee for pointing this out. **We used "type of surface" instead of "soil" when referring to snow, firn and ice in the manuscript**.

L210: in modeling parlance, "true" has a very specific meaning. Please revise.

R. We thank the referee for pointing this out. **We changed "true" to "actual" in L255**.

L218: See general comments regarding the detail that is needed about the SMB model calibration process. Also, why do you compare the 2000-15 simulation with the 2000-19 Minowa estimates? Is it not possible to compare a common period?

R. Please see our response to the general comments. Regarding the periods of comparison, Minowa et al. estimates were informed only for the full 2000-19 period. Consequently, we are only capable of comparing our results against that period.

L229: These biases could also result from inadequacies in the CR2MET product. In particular, if it is station-based, previous research has shown that meteorological station data in Patagonia is unreliable, particularly precipitation.

R. Please our reply to comment on L132. We are aware of potential inadequacies in the CR2MET product.

L231: a similar analysis could be performed by perturbing some of the model parameters (see general comments).

R. Please see our reply to general comments.

L260: If I understand correctly, AAO was calculated only for the 1979-2000 period? But SMB is available until 2015? Maybe it'd be useful to have a summary table with all datasets used, indicating time window, time-step, and citation.

R. We used the AAO data computed for every day from 1980-01-01 to 2015-12-31. The daily value of the AAO can be calculated for any date, projecting the daily height anomalies at 700 hPa poleward 20º S onto a particular atmospheric pattern. To derive that pattern (the leading mode of EOF analysis of monthly mean 700 hPa height) it is considered only the data between 1979 and 2000. **We clarify this point in the revised version of the manuscript (L182-184)**. However, we decided not to include a summary table in order not to increase the length of the manuscript.

L272: please reword to remind the reader that all these numerical quantities are estimates from your model. Also, the fact that annual SMB is positive means that for the ice fields to be in equilibrium (or decreasing in mass, as the literature suggests) then calving should account for the excess mass. Is that right? Also: there appears to be a slight increasing trend in the simulated SMB? Could you comment on this?

R. We thank the referee for this comment. 1. **We reformulated the paragraph as suggested (L329-339)**. 2. Yes, calving accounts for the excess of mass. **We referred to this in the Introduction (L98-99)**. 3. Precisely, there is a slight increasing trend in the simulated SMB, associated with a positive trend in the simulated precipitation. Nonetheless, in this investigation, we decided not to focus on the trend of the variables in order to maintain the purpose of the investigation delimited.

L298: How do you interpret the fact that although insolation shows the second-highest correlation with annual SMB (line 287), then the local-scale control indicates exactly the opposite? Maybe I'm missing something, but these two results seem inconsistent. Please clarify. Is this result sensitive to assumptions regarding the seasonal evolution of snow, firn and ice albedo? Nevertheless, it is expected that, unlike glaciers in mediterranean regions, solar radiation should have a minor role compared to temperature in Patagonia. High relative humidity and high very persistent cloud cover are coherent with this result.

R. We thank the referee for this comment. The fact that insolation shows the second-highest correlation with annual SMB (r=-0.44*) does not necessarily imply that annual variations of insolation exert an effective control over the annual variation in SMB. We interpret the correlation between insolation and SMB as a by-product of the correlation between precipitation and SMB. The sensitivity analysis assesses the causes of the year-to-year variations of the SMB, while the correlation analysis does not assess causality. **We devoted a paragraph in the discussion explaining this point (please see L543-555)** and gave an example.

L406: a small detail: probably "good" is not an appropriate adjective for describing correlation.

R. We thank the referee for pointing this out. **We replaced "good" with "statistically significant" in the manuscript (L483)**.

L446: suggest replacing "maintains" with "remains".

R. Done.

Response to anonymous referee #3

The referee's comments are in blue, our response in normal text, and the specifics actions are highlighted in bold.

General comments

The study analyzes the control of the present-day climate on the surface mass balance of the Patagonian Icefields. The main goals of the study are clearly formulated, and the study is well structured and written over largest parts. In the Discussion section, a stronger comparison with and discussion of results of other SMB studies in the region could strengthen the findings. Overall, the study adds valuable knowledge to the understanding of the interaction between climate and glacier mass balance in the southern Andes. I have one major comment which needs addressing and revision before publishing the article, together with several minor comments.

Major comment

The main limitation of this study is the fact that there is no spatial analysis for the correlation of SMB with large-scale indices and climate. It is possible that e.g. SAM does have an important impact on the SMB of the southern SPI, however, not on the whole study site. Averaging over such a large area can cause different signals in different regions to equal out. The study site does stretch over a large latitudinal band, and we know from literature that the climate and glaciology of NPI and SPI can show different characteristics and patterns. This is taken into account by calibrating the SMB model for both icefields individually, but then ignored throughout the rest of the paper.

Overall, I think by the spatial averaging a lot of valuable information is lost. I advise to conduct a spatiotemporal analysis instead of averaging over the SPI and NPI in order to gain information about the regional variability of climatic control on the SMB in Patagonia.

R. We thank the referee for this valuable comment. In this revised version of the manuscript, **we conducted a spatiotemporal analysis on the SMB field using Empirical Orthogonal Function (EOF) analysis** (Sect 3.1 and Figs. S6 and S7). We found that the annual time series of the spatially averaged SMB used in this work virtually coincides with the leading mode of the interannual variability of the SMB, which explains 59% of the total variance, dominating most of the area of the Patagonian Icefields. Thus, our results are representative of the majority of the Patagonian Icefields' area.

Furthermore, we provide a separate analysis for the NPI and the SPI (Sect. 3.1 and Figs. S8 and S9). We computed the NPI-only and SPI-only annual SMB time series and repeated the analysis from Sect. 3.5 for each series independently. The separate analysis confirms our main results, yet, a more inhibited tropical signal in the climatic control of the NPI-only annual SMB is evident compared to the SPI-only annual SMB.

Minor comments

L1: "Northern and Southern Patagonian Icefields" should be "Northern and Southern Patagonian Icefield"

R. Done.

L56 & 76 & 482: "dryer" should be "drier"

R. Done.

L88: The word "scenario" is strongly associated with climate scenarios, I recommend to reformulate

R. We thank the referee for this recommendation. **We reformulated the sentence in the revised version of the manuscript (please see L89)**.

L93: "assess" to "assesses"

R. Done

L112-123: The paragraph about the study site is a bit short in my opinion. I would include some brief information about major differences between the two icefields (e.g., SPI many marine terminating glaciers; do we have substantial climatic differences between the two icefields?). A reference to Fig. 2a would make sense here.

R. We thank the referee for this suggestion. We decided not to extend more the section about the study area in order not to increase the length of the manuscript. In our opinion, the literature mentioned in Sect. 2 contains sufficient information about the study area which can be revised by an interested reader.

L125-131: Which exact variables are taken form RecCMv4?

R. We took the near-surface air temperature, precipitation, and surface downward solar radiation fields from the RegCMv4. **We included this information explicitly in L133**.

L133: Why are two different versions used for precipitation and temperature?

R. We used the last versions available of the CR2MET products of precipitation and temperature. As these products are independent, there is no problem in using different versions for each variable.

L132-140: Both datasets, RefCMv4 and CR2MET, are (at least partly) based on ERA-Interim. I miss a comparison with an independent dataset. What about weather station data, or an independent Reanalysis dataset?

R. In this revised version of the manuscript, we compared annual precipitation and temperature estimates against available weather station measurements in Patagonia (but outside the Patagonian icefields' area) during 1980-2015 (**please see Sect. 2.2 and Figs. S2-S4**). CR2MET shows a slight cold and wet bias, but it reproduces the interannual variability of the available time series with high accuracy.

L150: You used the abbreviation ERA-Interim before. Introduce it at the first mentioning, please.

R. Done.

L154: Dot is missing at the end of the sentence.

R. Fixed

L158: This is not clear to me: "Lastly, we spatially unweighted averaged the meteorological forcing and the glaciological over the Patagonian Icefields…"

R. We thank the referee for warning us about this unclear expression. In this step we computed the spatial average of the meteorological forcing assigning the same weight to each grid point. **We clarified this in the revised version of the manuscript (please see L195-197).**

L164: The first "DEM" can be removed.

R. Done.

L183f.: This is not a downscaling of radiation, but simple interpolation.

R. We thank the referee for pointing this out. Now we refer to our treatment of the solar radiation as "bilinear interpolation" throughout the manuscript.

L199: I would replace the 10800s in the equation by a variable representing the timestep

R. Done

L201: These are not soil. Rather call it type of surface.

R. We thank the referee for pointing this out. We used "type of surface" instead of "soil" when referring to snow, firn and ice throughout the manuscript.

L208: Accurately, the end of summer season would be the 31 March.

R. We thank the referee for pointing this out. **We referred to April 1st as the start of the autumn season (L252).**

L218: Please, use a consistent number of decimal places.

R. Done.

L221-223: The values for c0 are very different between NPI and SPI. Why is this the case?

R. These values reflect the difference in SMB between NPI and SPI having similar sensitivity to near surface temperature.

L227: See comment to L132-140.

R. Please see our response to the minor comment on L132-140.

L241 & Table1: I recommend using a different abbreviation for the time period here to avoid confusion, as T has been used for temperature before.

R. Done. **We used Greek letter Tau for the time period in the revised version of the manuscript**.

Table 1 and following tables: It is common to put the table captions above the respective table.

R. We thank the referee for this recommendation. **We put the table captions above the respective table in the revised version of the manuscript**.

Table 2: The annual SMB and precipitation value does not exactly add up from winter and summer values. Rounding error?

R. Yes, winter and summer value do not add exactly in the case of annual SMB and precipitation because of rounding errors.

L286-289: Only mention the significant correlations here: "Among the modeled meteorological variables, the annual SMB is found to have the largest correlation with the annual precipitation (r = 0.69), followed by annual insolation (r = −0.44) (see Table 3). The same order is also evident in winter. The correlation between the SMB and temperature is only significant in summer."

We thank the referee for this recommendation. **We modified the paragraph as suggested (please see L345-348).**

L332: The correlation seems to be highest especially over the SPI?

R. While the map does not show correlation directly, effectively it shows the highest slope of regression over the SPI.

L346: "shows" to "show"

R. Done.

Fig. 6b: The grey and white shading is confusing at the first glance, as it seems like there would be two different variables in this plot like it is in panel d. Maybe you can give the shading the same color as the contours to make it clearer.

R. We thank the referee for this suggestion. Considering also suggestions from Referee #1, **we decided to use segmented black contours and grey shading for negative anomalies and normal black contours and white shading for positive anomalies (Fig. 5-7).**

L368f.: Refer to Fig. 9a here first.

R. Done.

L392ff.: The low correlation with the ENSO and SAM could be due to the spatial averaging over the whole study site. Consider differentiating into regions.

R. Please see our reply to the major comment. Additionally, we included in the revised version of the manuscript a section devoted to analyze independently the climatic control over the NPI-only and SPI-only annual SMB time series (**please see Sect. 3.7 and Figs. S8-S9**).

L418-426: Discussion and comparison with other SMB studies in southern Patagonia would support your findings. Similar findings have been found before, e.g., at Grey and Tynall Glacier (Weidemann 2018, https://doi.org/10.3389/feart.2018.00081)

R. We thank the referee for this suggestion. We discussed and compared with other SMB studies as suggested (**please see L522-531**).

L473 & 490: "SBM" to "SMB"

R. Done.

L476-489: Every paragraph starts with "years of … SMB are characterized by …". Consider reformulating.

R. We thank the referee for this suggestion. **We reworded the paragraphs as suggested (please see L603-619).**